# On the Limitations of Fractal Dimension
# as a Measure of Generalization

**Charlie B. Tan**[*]
University of Oxford

**Inés García-Redondo**[*]
Imperial College London

**Qiquan Wang**[*]
Imperial College London

**Michael M. Bronstein**
University of Oxford / Aithyra

**Anthea Monod**
Imperial College London

## Abstract

Bounding and predicting the generalization gap of overparameterized neural networks remains a central open problem in theoretical machine learning. There is a recent and growing body of literature that proposes the framework of fractals to model optimization trajectories of neural networks, motivating generalization bounds and measures based on the fractal dimension of the trajectory. Notably, the persistent homology dimension has been proposed to correlate with the generalization gap. This paper performs an empirical evaluation of these persistent homology-based generalization measures, with an in-depth statistical analysis. Our study reveals confounding effects in the observed correlation between generalization and topological measures due to the variation of hyperparameters. We also observe that fractal dimension fails to predict generalization of models trained from poor initializations. We lastly reveal the intriguing manifestation of model-wise double descent in these topological generalization measures. Our work forms a basis for a deeper investigation of the causal relationships between fractal geometry, topological data analysis, and neural network optimization.

## 1 Introduction

Deep learning enjoys widespread empirical success despite limited theoretical support. Measures from statistical learning theory, such as Rademacher complexity [Bartlett and Mendelson, 2002] and VC-Dimension [Hastie et al., 2009], indicate that without explicit regularization, over-parameterized models will generalize poorly. In contrast, neural networks are able to generalize strongly despite having sufficient capacity to simply memorize their training data [Liu et al., 2020, Zhang et al., 2021]. Remarkably, neural networks often exhibit improved generalization for increases in capacity [Nakkiran et al., 2021]. Describing the generalization behavior of neural networks therefore requires the development of novel learning theory [Zhang et al., 2021]. Ultimately, deep learning theory seeks to define generalization bounds for given experimental configurations [Valle-Pérez and Louis, 2020], such that the generalization error of a given model can be bounded and predicted [Jiang et al., 2020].

Generalization is typically attributed to the implicit bias of gradient-based optimization. A number of works have considered the geometry of generalizing solutions within parameter space [Dinh et al., 2017, Garipov et al., 2018], and the bias of optimization methods towards such solutions [He et al., 2019, Izmailov et al., 2018]. Şimşekli et al. [2020] propose *random fractal* structure for neural network optimization trajectories and compute generalization bounds based on *fractal dimensions*. However, this work requires rigid topological and statistical conditions on the optimization trajectory as well as the learning algorithm. Subsequent work by Birdal et al. [2021] proposes the use of

---

[*]Equal contribution. Correspondence to: charlie.tan@cs.ox.ac.uk, i.garcia-redondo22@imperial.ac.uk, qiquan.wang17@imperial.ac.uk, michael.bronstein@cs.ox.ac.uk, a.monod@imperial.ac.uk. Code provided for all experiments at: `https://github.com/charliebtan/fractal_dimensions`

38th Conference on Neural Information Processing Systems (NeurIPS 2024).

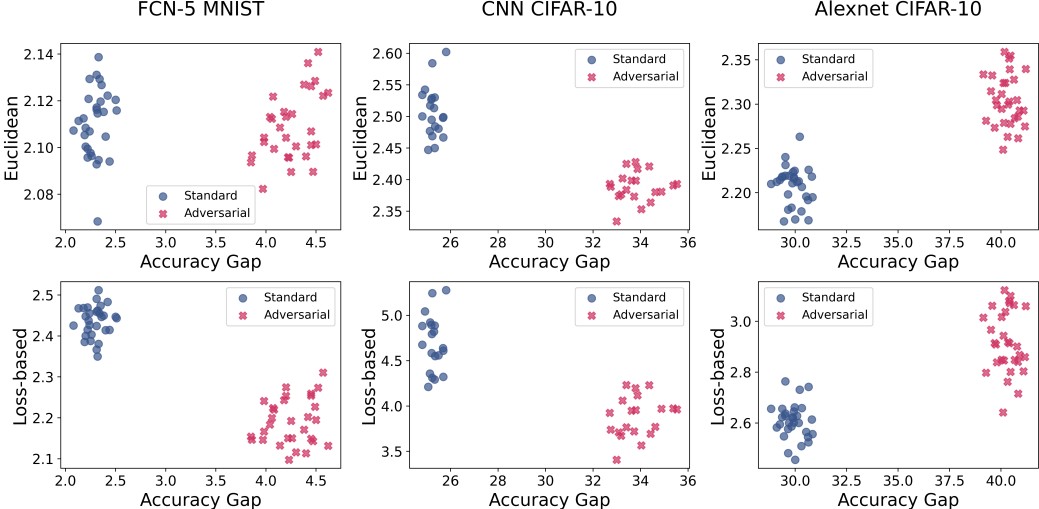

Figure 1: **Adversarial initialization is a failure mode for PH dimension-based generalization measures**. Training models from an adversarial initialization leads to higher accuracy gap than for models trained from random initialization. Both PH dimensions fail to correctly attribute higher values to the poorly generalizing models on FCN-5 MNIST and CNN CIFAR-10.

*persistent homology (PH) dimension* [Adams et al., 2020]—a measure of fractal dimension deriving from topological data analysis (TDA)—to relax these assumptions. They propose an efficient procedure for estimating PH dimension, and apply this both as a measure of generalization and as a scheme for explicit regularization. Dupuis et al. [2023] extend this approach using a data-dependent pseudometric to further relax continuity assumptions on the network loss.

Our paper constitutes an extended empirical evaluation of the performance and viability of these proposed topological measures of generalization; in particular, robustness and failure modes are explored in a wider range of experiments than those considered by Birdal et al. [2021] and Dupuis et al. [2023]. Our main contributions are as follows:

- We reproduce the learning rate/batch size grid experiments of Dupuis et al. [2023], observing comparable correlation coefficients, aside from the loss-based PH dimension on AlexNet CIFAR-10 where models trained with high learning rate are attributed high dimension values despite having small generalization gap.
- We extend the statistical analysis of Dupuis et al. [2023] to include partial correlations. We observe that in some cases learning rate has a significant influence on observed correlation between PH dimensions and generalization gap for fixed batch sizes.
- We further conduct a conditional independence test using the conditional mutual information [Jiang et al., 2020], observing that both Euclidean and loss-based PH dimension are conditionally independent of generalization gap on MNIST.
- We train models of varying generalization gap using adversarial initialization [Liu et al., 2020]. As presented in Figure 1, we observe that both dimensions fail to correctly attribute high values to poorly generalizing models for some architectures and datasets.
- We train a CNN architecture at a range of width multipliers to reproduce the model-wise double descent of Nakkiran et al. [2021]. Neither PH dimension correctly correlates with generalization gap in this setting. Interestingly, by correlating with test accuracy, double descent manifests in Euclidean PH dimension.

## 2   Background

Following Dupuis et al. [2023], let $(\mathcal{Z}, \mathcal{F}_{\mathcal{Z}}, \mu_{\mathcal{Z}})$ be the data space, where $\mathcal{Z} = \mathcal{X} \times \mathcal{Y}$, and $\mathcal{X}, \mathcal{Y}$ represent the feature and label spaces respectively. We aim to learn a parametric approximation $h_w : \mathcal{X} \times \mathcal{W} \to \mathcal{Y}$ of the unknown data generating distribution $\mu_{\mathcal{Z}}$ from a finite set of i.i.d. training points $S := \{z_1, \ldots, z_n\} \sim \mu_{\mathcal{Z}}^{\otimes n}$. The quality of our parametric approximation is measured using a loss function $\mathcal{L} : \mathcal{Y} \times \mathcal{Y} \to \mathbb{R}$ composed with the parametric approximation $\ell(\omega, z) := \mathcal{L}(h_\omega(x), y)$.

The learning task then amounts to solving an optimization problem over parameters $w \in \mathbb{R}^d$, where we seek to minimize the *empirical risk* $\hat{\mathcal{R}}(w, S) := \frac{1}{n} \sum_{i=1}^{n} \ell(w, z_i)$ over a finite set of training points. To measure performance on unseen data samples, we consider the *population risk* $\mathcal{R}(w) := \mathbb{E}_z[\ell(w, z)]$ and define the *generalization gap* to be the difference of the population and empirical risks $\mathcal{G}(S, w) := |\mathcal{R}(w) - \hat{\mathcal{R}}(S, w)|$. For a given training dataset and some initial value for the weights $w_0 \in \mathbb{R}^d$ we refer to the optimization trajectory as $\mathcal{W}_S$.

## 2.1 Persistent Homology and Fractal Dimension

Fractals arise in recursive processes [Mandelbrot, 1983, Prähofer and Spohn, 2000]; chaotic dynamical systems [Briggs, 1992, Mandelbrot et al., 2004]; and real-world data [Mandelbrot, 1967, Coleman and Pietronero, 1992, Falconer, 2007, Pietronero and Tosatti, 2012]. A key characteristic is their *fractal dimension*, first introduced as the *Hausdorff dimension* [Hausdorff, 1918]. Due to its computational complexity, more efficient measures such as the *box-counting dimension* Sarkar and Chaudhuri [1994] were later developed. An alternative fractal dimension can also be defined in terms of *minimal spanning trees* of finite metric spaces [Kozma et al., 2006]. A recent line of work by Adams et al. [2020] and Schweinhart [2021, 2020] extended and reinterpreted fractal dimension using PH. Originating from algebraic topology, PH provides a systematic framework for capturing and quantifying the multi-scale topological features in complex datasets through a topological summary called the *persistence diagram* or *persistence barcode*; further details on PH are given in Appendix A and a more comprehensive exposition of fractal dimension can be found in Appendix B.

We follow the approach by Schweinhart [2021] to define a PH-based fractal dimension. Let $\mathbf{x} = \{x_1, \ldots, x_n\}$ be a finite subset of a metric space $(X, \rho)$. Let $\mathrm{PH}_i(\mathbf{x})$ be the persistence diagram obtained from the PH of dimension $i$ computed from the Vietoris–Rips filtration and define

$$E_\alpha^i(\mathbf{x}) := \sum_{(b,d) \in \mathrm{PH}_i(\mathbf{x})} (d - b)^\alpha. \tag{1}$$

**Definition 2.1** ([Schweinhart, 2020]). Let $S$ be a bounded subset of a metric space $(X, \rho)$. The *ith PH dimension* (PH$_i$-dim) for the Vietoris–Rips complex of $S$ is

$$\dim_{\mathrm{PH}}{}^i(S) := \inf \left\{ \alpha : \exists\, C > 0 \text{ s.t. } E_\alpha^i(\mathbf{x}) < C, \, \forall\, \mathbf{x} \subset S \text{ finite subset} \right\}.$$

Fractal dimensions need not be well-defined for all subsets of a metric space. However, under a certain regularity condition (*Ahlfors regularity*), the Hausdorff and box-counting dimensions are well defined and coincide [Falconer, 2007]. Additionally, for any metric space, the minimal spanning tree is equal to the upper box dimension [Kozma et al., 2006]. The relevance of PH appears when considering the minimal spanning tree in fractal dimensions. Specifically, there is a bijection between the edges of the Euclidean minimal spanning tree of a finite metric space $\mathbf{x} = \{x_1, \ldots, x_n\}$ and the points in the persistence diagram $\mathrm{PH}_0(\mathbf{x})$ obtained from the Vietoris–Rips filtration. This automatically gives the equivalence $\dim_{\mathrm{PH}}{}^0(S) = \dim_{\mathrm{MST}}(S) = \dim_{\mathrm{box}}(S)$.

## 2.2 Fractal Dimension-Based Generalization Bounds

Şimşekli et al. [2020] empirically observe that the gradient noise exhibits heavy-tailed behavior, which they use to model stochastic gradient descent (SGD) as a discretization of a *decomposable Feller process*. They also impose initialization with zeros; that $\ell$ is bounded and Lipschitz continuous in $w$; and that $\mathcal{W}_S$ is bounded and Ahlfors regular. In this setting, they compute two bounds for the worst-case generalization error, $\max_{w \in \mathcal{W}_S} \mathcal{G}(S, w)$, in terms of the Hausdorff dimension of $\mathcal{W}_S$. They first prove bounds related to covering numbers (used to define the upper-box counting dimension) and then use Ahlfors regularity to link the bounds to the Hausdorff dimension.

Subsequently, Birdal et al. [2021] further develop the bounds in Şimşekli et al. [2020] by reformulating them in terms of the 0-dimensional PH dimension [Schweinhart, 2021] of $\mathcal{W}_S$. The link between the upper-box dimension and the 0-dimensional PH dimension, which is the cornerstone of their proof, only requires boundedness of $\mathcal{W}_S$ (which is also one of the assumptions by Şimşekli et al. [2020]), thus eliminating the Ahlfors regularity condition. In order to estimate the PH dimension, they prove (see Proposition 2, [Birdal et al., 2021]) that for all $\epsilon > 0$ there exists a constant $D_{\alpha,\epsilon}$ such that

$$E_\alpha^0(W_n) \leq D_{\alpha,\epsilon}\, n^\beta, \tag{2}$$

where $\beta := \frac{\dim_{\mathrm{PH}}{}^0(\mathcal{W}_S)+\epsilon-\alpha}{\dim_{\mathrm{PH}}{}^0(\mathcal{W}_S)+\epsilon}$ for all $n \geq 0$, all i.i.d. samples $W_n$ with $n$ points on the optimization trajectory $\mathcal{W}_s$, and $E_\alpha^0(W_n)$ as defined in (7). Using this result, they estimate and bound the PH-dimension by fitting a power law to the pairs $(\log(n), \log(E_1^0(W_n)))$. They then use (2) to estimate $\dim_{\mathrm{PH}}{}^0(\mathcal{W}_s) \approx \frac{\alpha}{1-m}$, where $m$ is the slope of the regression line. Concurrently, Camuto et al. [2021] take a different, non-topological approach by studying stationary distributions of Markov chains and describing the optimization algorithm as an iterated function system (IFS). They establish generalization bounds with respect to the upper Hausdorff dimension of a limiting measure. Most recently, Dupuis et al. [2023] further develop the topological approach by Birdal et al. [2021] to circumvent the Lipschitz condition on the loss function that was required in all previous works by obtaining a bound depending on a data-driven pseudometric $\rho_S$ instead of the $\mathbb{R}^d$ Euclidean metric,

$$\rho_S(w, w') := \frac{1}{n}\sum_{i=1}^n |\ell(w, z_i) - \ell(w', z_i)|, \quad \forall \, w, \, w' \in \mathbb{R}^d. \tag{3}$$

They derive bounds for the worst-case generalization gap, where the only assumption is that the loss $\ell : \mathbb{R}^d \times \mathcal{Z} \to \mathbb{R}$ is continuous in both variables and uniformly bounded by some $B > 0$. These bounds are established with respect to the upper-box dimension of the set $\mathcal{W}_S$ using $\rho_S$ (see Theorem 3.9. [Dupuis et al., 2023]). They additionally prove that for pseudometric-bounded spaces, the corresponding upper-box counting dimension coincides with the $0$-dimensional PH-dimension, which they estimate as in Birdal et al. [2021].

## 3  Experimental Setup

Our experiments closely follow the setting of Dupuis et al. [2023]. We train with SGD until convergence, then continue for 5000 additional iterations to obtain a sample optimization trajectory $\{w_k : 0 < k \leq 5000\}$ about the local minimum attained. We then compute the $0$-PH dimension using both the Euclidean metric and the loss-based pseudometric (3) to obtain the generalization measures of Birdal et al. [2021] and Dupuis et al. [2023]. In keeping with the assumptions of this theory, we omit explicit regularization such as dropout or weight decay, and maintain constant learning rates in all experiments. Following Dupuis et al. [2023] we define the generalization gap as the the absolute accuracy gap for classification tasks, and the absolute loss gap in regression tasks. Further details on the experimental setup, expanding on this section, are provided in Appendix C and a note on the stability of PH dimension estimates post-convergence can be found in Appendix D.

**Datasets and Architectures.** We employ the same datasets and architectures as Dupuis et al. [2023]: (i) fully-connected network of 5 (FCN-5) and 7 (FCN-7) layers on the California housing dataset (CHD) [Kelley Pace and Barry, 1997]; (ii) FCN-5 and FCN-7 on the MNIST dataset [Lecun et al., 1998]; and (iii) AlexNet [Krizhevsky et al., 2017] on the CIFAR-10 dataset [Krizhevsky, 2009]. We additionally conduct experiments on a 5-layer convolutional neural network, defined by Nakkiran et al. [2021] as *standard CNN*, trained on CIFAR-10 and CIFAR-100, with batch normalization removed to align with the bound assumptions.

**Overview of Experiments.** We divide our experiments into three categories. In the first, we replicate the $6 \times 6$ grid of learning rates and batch sizes considered in Dupuis et al. [2023]. We use these results to perform a statistical analysis of the correlation between PH dimension (both Euclidean and loss-based) and generalization error, particularly exploring the influence of the hyperparameters. Second, we use *adversarial pre-training* as a method to generate poorly generalizing models [Liu et al., 2020]. Finally, we consider model-wise double descent, the phenomenon in which test accuracy is non-monotonic with respect to increasing model parameters [Nakkiran et al., 2021].

**Computational Resources.** All experiments were run on high performance computing clusters using GPU nodes with Quadro RTX 6000 (128 CPU cores) or NVIDIA H100 (192 CPU cores). The runtime for training and computing the PH dimension vary for different architectures, datasets, and hardware used, with the longest experiments taking several hours.

## 4  Grid Experiments and Correlations

We first reproduce the experiments of Dupuis et al. [2023], wherein learning rate and batch size are defined in a $6 \times 6$ grid as defined in Appendix C. In Figure 2 we present the results of these

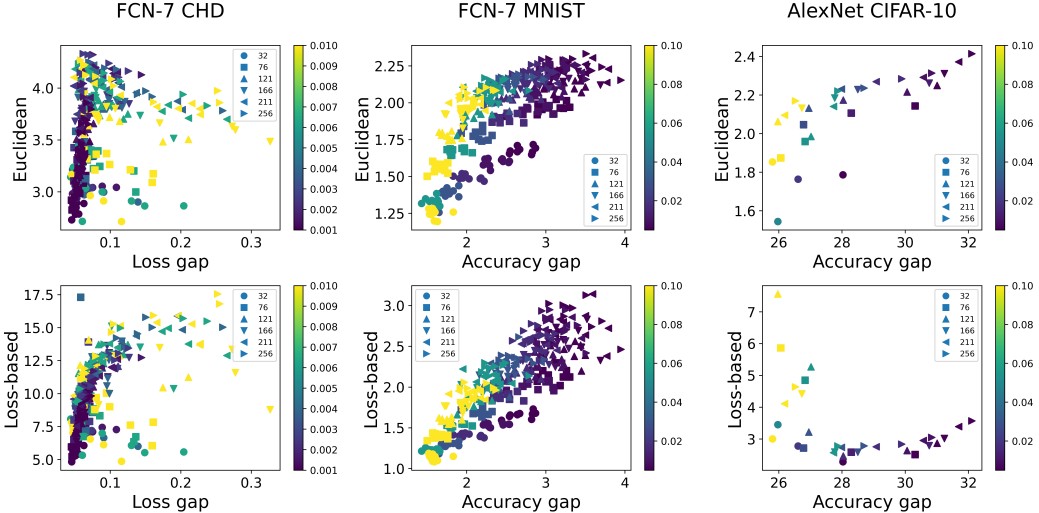

**Figure 2: Learning rate/batch size grid results.** Euclidean (top) and loss-based (bottom) PH dimension plotted against generalization gap for range of learning rates and batch sizes.

experiments for the PH dimensions in the FCN-7 and AlexNet experiments, additional results for the FCN-5 models and other generalization measures are presented in Appendix E.

## 4.1 Correlation Analysis

In Table 1, we present correlation coefficients between the PH dimensions (Euclidean and loss-based) and generalization gap. As in Dupuis et al. [2023], we present Spearman's rank correlation coefficient $\rho$; the mean granulated Kendall rank correlation coefficient $\Psi$ [Jiang et al., 2020]; and the standard Kendall rank correlation coefficient $\tau$. We also compute correlations with the $\ell^2$ norm of the final parameter vector $\|w_{5000}\|_2$, and the learning rate/batch size ratio. We emphasize the learning rate/batch size ratio is not a measure of generalization as it is not a measurable experimental output.

The results on CHD and MNIST align with those reported by Dupuis et al. [2023]: both the Euclidean and loss-based measures have positive correlation with generalization gap, with the correlation of the loss-based being slightly stronger than that of the Euclidean. However, for the AlexNet CIFAR-10 experiment, we obtain negative correlations for the loss-based measure, in contrast to the theory and results of Dupuis et al. [2023]. Observing the results in Figure 2, we see this is due to several points with high learning rate achieving very high PH dimension. We are unable to determine why these points appear in our experiments but not prior studies. However, we assert that all points considered attain 100% training accuracy hence meet the convergence assumption of Dupuis et al. [2023].

The $\ell^2$ norm has the strongest absolute correlations for all experiments, but this correlation is positive for regression and negative for classification. The positive correlation on regression experiments is unexpected, although similar behavior has been observed by Jiang et al. [2020]. The learning rate/batch size ratio has strong negative correlation on classification experiments and weak positive correlation on the regression experiments. The strong correlation of learning rate/batch size ratio on classification experiments aligns with trends observable in Figure 2, indicating potential confounding effects of these variables in the observed correlations. Dupuis et al. [2023] included the mean granulated Kendall rank correlation coefficient $\Psi$ in their analysis to mitigate the influence of hyperparameters when computing rank correlations. This coefficient is computed by taking the average over Kendall coefficients at fixed values of the hyperparameters. However, by averaging over all hyperparameter ranges, significant correlations for different fixed values of the hyperparameters might be masked by lower correlations, resulting in inconclusive findings.

## 4.2 Partial Correlation

Given the correlation between learning rate/batch size ratio and generalization gap, we study partial correlations to isolate the influence of these hyperparameters on the observed correlation between generalization and PH dimensions. Suppose $X$ and $Y$ are our variables of interest and $Z$ is a

Table 1: Spearman's $\rho$, mean granulated Kendall $\Psi$ and Kendall $\tau$ rank coefficients of the correlation with generalization gap. For CHD and MNIST mean of 10 seeds presented with standard deviation.

| | Measure | $\rho$ | $\Psi$ | $\tau$ |
|---|---|---|---|---|
| FCN-5 CHD | Euclidean | $0.71_{\pm0.07}$ | $0.48_{\pm0.07}$ | $0.54_{\pm0.07}$ |
| | Loss-based | $0.78_{\pm0.06}$ | $0.64_{\pm0.06}$ | $0.64_{\pm0.06}$ |
| | Norm | $0.92_{\pm0.04}$ | $0.84_{\pm0.05}$ | $0.81_{\pm0.06}$ |
| | LR / BS | $0.29_{\pm0.11}$ | $0.11_{\pm0.08}$ | $0.21_{\pm0.07}$ |
| FCN-7 CHD | Euclidean | $0.45_{\pm0.07}$ | $0.19_{\pm0.07}$ | $0.32_{\pm0.06}$ |
| | Loss-based | $0.67_{\pm0.11}$ | $0.51_{\pm0.09}$ | $0.54_{\pm0.08}$ |
| | Norm | $0.87_{\pm0.05}$ | $0.78_{\pm0.05}$ | $0.72_{\pm0.06}$ |
| | LR / BS | $0.15_{\pm0.01}$ | $0.04_{\pm0.05}$ | $0.10_{\pm0.09}$ |
| FCN-5 MNIST | Euclidean | $0.67_{\pm0.05}$ | $0.73_{\pm0.05}$ | $0.50_{\pm0.04}$ |
| | Loss-based | $0.77_{\pm0.05}$ | $0.79_{\pm0.05}$ | $0.60_{\pm0.04}$ |
| | Norm | $-0.93_{\pm0.03}$ | $-0.79_{\pm0.04}$ | $-0.81_{\pm0.04}$ |
| | LB / BS | $-0.95_{\pm0.02}$ | $-0.83_{\pm0.05}$ | $-0.84_{\pm0.04}$ |
| FCN-7 MNIST | Euclidean | $0.78_{\pm0.04}$ | $0.88_{\pm0.04}$ | $0.61_{\pm0.04}$ |
| | Loss-based | $0.88_{\pm0.03}$ | $0.90_{\pm0.03}$ | $0.71_{\pm0.04}$ |
| | Norm | $-0.97_{\pm0.01}$ | $-0.83_{\pm0.05}$ | $-0.88_{\pm0.04}$ |
| | LR / BS | $-0.98_{\pm0.00}$ | $-0.91_{\pm0.03}$ | $-0.90_{\pm0.02}$ |
| AlexNet CIFAR-10 | Euclidean | $0.85$ | $0.85$ | $0.69$ |
| | Loss-based | $-0.31$ | $-0.07$ | $-0.14$ |
| | Norm | $-0.98$ | $-0.94$ | $-0.91$ |
| | LR / BS | $-0.98$ | $-0.94$ | $-0.91$ |

multivariate variable. The partial correlation between $X$ and $Y$ given $Z$ is the correlation between the residuals of the regressions of $X$ with $Z$ and of $Y$ with $Z$. If the correlation between $X$ and $Y$ can be fully explained by $Z$, then the partial correlation should yield a low coefficient. To report statistical significance, we conduct a non-parametric permutation-type hypothesis test for the assumption that the partial correlation is equal to zero. In our setting, the null hypothesis implies the correlation observed between generalization and PH dimensions is explained by the influence of other hyperparameters.

In Table 2, we report the partial Spearman's and (standard) Kendall rank correlation between generalization gap and PH dimensions, conditioned on learning rate for fixed batch sizes. We provide the corresponding $p$-values for the stated hypothesis test in parentheses. Recall that a $p$-value lower than 0.05 implies the rejection of the null hypothesis, or equivalently, that there is a correlation between PH dimension and generalization gap that cannot be explained by the influence of the hyperparameters; a $p$-value greater than 0.05 implies a significant influence of the corresponding hyperparameter in the apparent correlation. We observe for most batch sizes, the correlation present between Euclidean dimension and generalization gap can be explained by the influence of learning rate, particularly for larger batch sizes. There is no consistent trend for the loss-based dimension, and the influence of the learning rate is found to be significant in fewer cases, indicating it may be a better-suited measure.

## 4.3 Conditional Independence

We have established that for some batch sizes, the correlation between PH dimension and generalization is significantly influenced by learning rate. We now seek to determine the existence of a causal relationship between the PH dimension and the generalization gap, basing our study on that of Jiang et al. [2020]. If a causal relationship does exist, then a low PH dimension caused by a variation of hyperparameters would consequently cause the generalization gap to be small. If a causal relationship does not exist, then the variation of the hyperparameter would cause both the PH dimension and generalization gap to be low, without any meaningful effect between the PH dimension and generalization gap. An illustrative example of these scenarios is provided in Figure 3.

Table 2: Partial Spearman's $\rho$ and Kendall $\tau$ correlation computed between PH dimensions and generalization error for fixed batch sizes given learning rate. $p$-values in parentheses; bolded entries have $p$-value $\geq 0.05$ signaling a significant influence of learning rate.

| | Batch size | Euclidean | | Loss-based | |
|---|---|---|---|---|---|
| | | $\rho$ | $\tau$ | $\rho$ | $\tau$ |
| FCN-5 CHD | 32 | 0.10 (**0.43**) | 0.06 (**0.48**) | 0.06 (**0.64**) | 0.04 (**0.66**) |
| | 65 | −0.03 (**0.85**) | −0.01 (**0.90**) | −0.10 (**0.47**) | −0.08 (**0.39**) |
| | 99 | −0.41 (0.00) | −0.29 (0.00) | −0.67 (0.00) | −0.49 (0.00) |
| | 132 | −0.31 (0.02) | −0.21 (0.02) | −0.65 (0.00) | −0.47 (0.00) |
| | 166 | −0.04 (**0.76**) | −0.02 (**0.79**) | −0.49 (0.00) | −0.33 (0.00) |
| | 200 | −0.05 (**0.70**) | −0.03 (**0.75**) | −0.65 (0.00) | −0.48 (0.00) |
| FCN-7 CHD | 32 | 0.48 (0.00) | 0.32 (0.00) | 0.37 (0.00) | 0.24 (0.01) |
| | 65 | 0.10 (**0.46**) | 0.07 (**0.42**) | −0.02 (**0.88**) | −0.02 (**0.86**) |
| | 99 | −0.35 (0.01) | −0.24 (0.01) | −0.73 (0.00) | −0.55 (0.00) |
| | 132 | 0.04 (**0.74**) | 0.02 (**0.87**) | −0.18 (**0.19**) | −0.14 (**0.13**) |
| | 166 | 0.08 (**0.56**) | 0.03 (**0.76**) | −0.70 (0.00) | −0.51 (0.00) |
| | 200 | 0.12 (**0.39**) | 0.08 (**0.37**) | −0.82 (0.00) | −0.66 (0.00) |
| FCN-5 MNIST | 32 | 0.63 (0.00) | 0.42 (0.00) | 0.46 (0.00) | 0.32 (0.00) |
| | 76 | −0.08 (**0.54**) | −0.06 (**0.51**) | 0.43 (0.00) | 0.29 (0.00) |
| | 121 | 0.17 (**0.21**) | 0.13 (**0.14**) | 0.37 (0.00) | 0.26 (0.00) |
| | 166 | 0.00 (**0.99**) | 0.01 (**0.95**) | 0.16 (**0.22**) | 0.12 (**0.18**) |
| | 211 | 0.22 (**0.10**) | 0.15 (**0.09**) | 0.17 (**0.20**) | 0.12 (**0.18**) |
| | 256 | 0.08 (**0.55**) | 0.07 (**0.48**) | 0.10 (**0.45**) | 0.09 (**0.34**) |
| FCN-7 MNIST | 32 | 0.81 (0.00) | 0.61 (0.00) | 0.82 (0.00) | 0.62 (0.00) |
| | 76 | 0.68 (0.00) | 0.46 (0.00) | 0.79 (0.00) | 0.58 (0.00) |
| | 121 | 0.29 (0.03) | 0.21 (0.02) | 0.69 (0.00) | 0.50 (0.00) |
| | 166 | 0.26 (**0.05**) | 0.17 (**0.05**) | 0.50 (0.00) | 0.34 (0.00) |
| | 211 | 0.26 (**0.46**) | 0.20 (0.03) | 0.45 (0.00) | 0.31 (0.00) |
| | 256 | 0.19 (**0.15**) | 0.16 (**0.07**) | 0.30 (0.02) | 0.21 (0.02) |

To distinguish between these two scenarios, we conduct a conditional independence test by computing the conditional mutual information (CMI) [Jiang et al., 2020] as defined in Appendix F. The CMI vanishes to zero if and only if $X \perp Y \mid Z$, i.e., $X$ (PH dimension) and $Y$ (generalization gap) are conditionally independent given $Z$ (hyperparameter). Given the discrete nature of our selected hyperparameters, we empirically determine the probability density functions. To assess the significance of the computed CMI, we generate a null distribution for the CMI under local permutations of $X$ or $Y$ for fixed hyperparameter values, where "local" here refers to the group of realizations of $X$ and $Y$ generated under the same hyperparameters $Z$ [Kim et al., 2022]. The null hypothesis implies that $X$ and $Y$ are conditionally independent, in which case the CMI is invariant to permutations. We reject the assumption of conditional independence if the CMI lies in the extremes of the null distribution.

Table 3 contains the results of the conditional independence test between PH dimensions and generalization conditioned on learning rate for fixed batch sizes. Within the table, a $p$-value $> 0.05$ implies the acceptance of the null hypothesis of conditional independence ($H_0$ in Figure 3), whereas a $p$-value $\leq 0.05$ indicates the existence of conditional dependence ($H_1$ in Figure 3). Hence, we observe that for the models trained on MNIST data and for most batch sizes, the PH dimensions and generalization can be considered to be conditionally independent. For the models trained on the CHD, for most batch sizes, the PH dimensions and generalization are seen to be conditionally dependent.

## 5 Adversarial Initialization

The theory of Birdal et al. [2021] and Dupuis et al. [2023] proposes a positive correlation between generalization and the respective PH dimensions. However, neither work makes an assumption on the initialization scheme applied a the start of training. To investigate the sensitivity of the measures to initialization, we employ the *adversarial initialization* technique proposed by Liu et al. [2020].

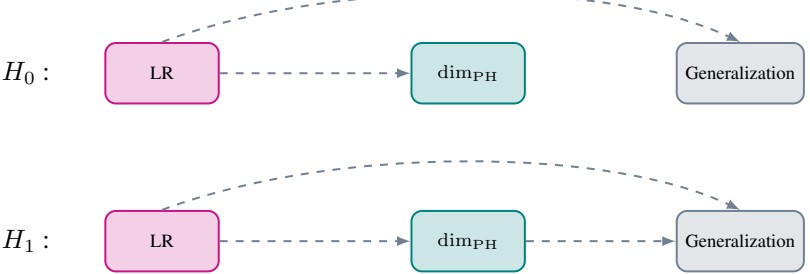

Figure 3: **Diagram of causal relationships under investigation in the conditional independence test.** In $H_0$ the PH dimension is conditionally independent of PH dimension given learning rate and there is no direct causal relationship between these variables. In $H_1$ generalization gap is conditionally dependent of the PH dimension indicating a causal relationship may exist.

Table 3: Table of $p$-values from conditional independence tests between PH dimensions and generalization gap conditioned on learning rate using conditional mutual information (CMI) as test statistic with local permutations for given batch sizes. Bolded $p$-values indicate conditional independence between PH dimension and generalization.

|  | **PH dimension** | **Batch size** | | | | | |
| --- | --- | --- | --- | --- | --- | --- | --- |
|  |  | 32 | 65 | 99 | 132 | 166 | 200 |
| FCN-5 CHD | Euclidean | 0.01 | **0.27** | 0.02 | 0.01 | 0.00 | **0.06** |
|  | Loss-based | 0.00 | 0.02 | 0.00 | 0.00 | 0.00 | 0.00 |
| FCN-7 CHD | Euclidean | 0.00 | **0.28** | 0.00 | 0.00 | 0.00 | 0.00 |
|  | Loss-based | 0.00 | **0.33** | 0.00 | 0.00 | 0.00 | 0.00 |
|  |  | **Batch size** | | | | | |
|  |  | 32 | 76 | 121 | 166 | 211 | 256 |
| FCN-5 MNIST | Euclidean | **0.18** | **0.57** | **0.35** | **0.11** | **0.18** | **0.40** |
|  | Loss-based | **0.23** | **0.28** | **0.07** | **0.09** | **0.11** | 0.01 |
| FCN-7 MNIST | Euclidean | **0.15** | 0.04 | **0.41** | **0.25** | **0.92** | **0.75** |
|  | Loss-based | 0.02 | 0.00 | **0.30** | **0.71** | **0.88** | **0.38** |

This entails a pre-training phase on training data with fixed, random labels until the network has successfully interpolated this random data. The resulting parameters are then used as initialization for training on the true dataset, leading to a poorly generalizing model.

In Figure 1, we present the results of this experiment, where the adversarial initialization models are contrasted against models trained from a standard (random) initialization. 30 seeds are evaluated for CHD and MNIST, and 20 seeds for AlexNet. We find that for the CNN CIFAR-10 and the FCN-5 MNIST the adversarial initialization models present lower PH dimensions despite having higher generalization gap, in contrast to the proposed theory. On AlexNet CIFAR-10 the PH dimensions both successfully identify the poorly generalization models, prescribing high values in this case.

## 6 Model-Wise Double Descent

We lastly explore model-wise double descent through the PH dimensions [Nakkiran et al., 2021]. In model-wise double descent the test accuracy of a classifier is non-monotonic with respect to the number of parameters—a surprising result in contrast with classical learning theory.

In Figure 4, we present results for the CNN trained on CIFAR-100 at a variety of width multipliers. We follow the "noiseless" configuration of Nakkiran et al. [2021], although we do not use batch normalization or learning rate decay to align with the assumptions of Dupuis et al. [2023]. The

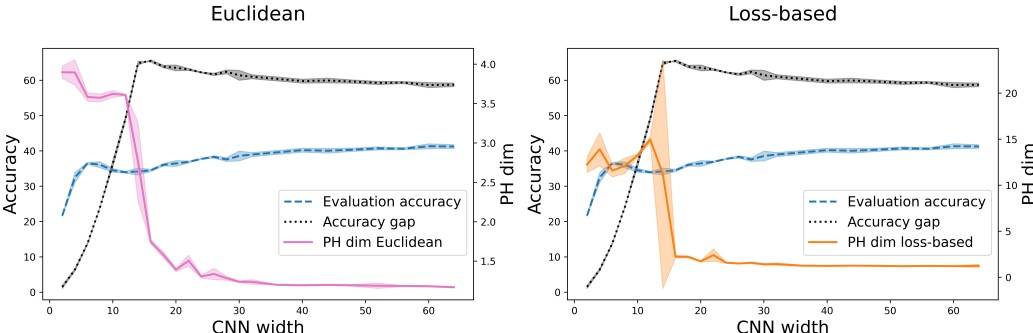

Figure 4: **Model-wise double descent manifests in Euclidean PH dimension, whilst neither PH dimension correlates with generalization gap in this setting.** Test accuracy, generalization gap, and PH dimensions for range of CNN widths. The double descent behavior is clearly visible in test accuracy and Euclidean PH dimension, but the generalization gap is monotonic in this critical region. Mean of three seeds with standard deviation shaded.

mean of 3 seeds is presented with standard deviation shaded. In evaluation accuracy, we observe the classical double descent behavior, but note that the generalization gap is monotonic in the range up to width multiplier of 16. We additionally observe a double descent behavior in Euclidean PH dimension. The behavior of loss-based PH dimension does not follow the double descent pattern, with notable instability and variance in the region of the evaluation accuracy double descent critical region. These results suggest a connection between the convergence properties of the model in the critical region of double descent with respect to the model width, itself a poorly understood phenomenon, and the Euclidean PH dimension. They also show the failure of either PH dimension to correlate with generalization gap for varying model width, which is monotonic in this width region.

## 7 Discussion

Our results demonstrate two modes of failure for PH dimensions as measures of generalization: adversarial initialization and model width variation in the critical region of double descent. Furthermore, we show that in some cases the correlation observed between PH dimension and generalization gap is significantly influenced by hyperparameter values. An explanation for the influence of hyperparameters—in particular learning rate—in the value of PH dimension is that the underlying space used in the PH computations is determined by samples from the optimization trajectory; and the influence of the learning rate on the geometry of these samples is notable. We further demonstrate that for some architectures and datasets the generalization measures and generalization gap are conditionally independent given the confounding hyperparameters, implying that the observed relationship between the two variables is not directly causal in these settings.

Evidently, the PH dimensions are not universally successful in correlating with generalization gap, in contrast to the general bounds proven by Birdal et al. [2021] and Dupuis et al. [2023]. We have two main conjectures for this disconnect between theory and practice. Firstly, the technical assumptions required by Dupuis et al. [2023] to prove generalization bounds, and their implications on architecture and hyperparameters valid for variation, are not clear. Preceding works [Şimşekli et al., 2020, Birdal et al., 2021] involve various technical assumptions about optimization trajectories and loss functions that may not be met in practice. Secondly, the term in the generalization bounds involving mutual information between the training data and the optimization trajectory may dominate, leading to vacuous bounds with respect to fractal dimension. These mutual information terms are complex and less studied than the fractal dimensions; it is unclear if they can be empirically estimated. The assumption that the mutual information does not dominate the bounds has not been proven and has been scarcely explored. We believe some experiments may enter a regime where this term dominates, disrupting the expected correlation.

### 7.1 Limitations

**Models and settings.** The goal of our study was to better understand the conclusions drawn by Birdal et al. [2021] and Dupuis et al. [2023]. We are thus subject to the following restrictions arising

from the assumptions in the theoretical results of Birdal et al. [2021] and Dupuis et al. [2023]: (i) we use only "vanilla" SGD; (ii) we only work with a constant learning rate; (iii) we do not use batch normalization; (iv) we do not study the addition of explicit regularization such as dropout or weight decay. We address this limitation by extending the study to include adversarial initialization scenarios [Liu et al., 2020], and studying the connection between double descent [Nakkiran et al., 2021] and PH dimension whilst still remaining within the theoretical assumptions. Future research directions include alternative optimization algorithms and common neural network architecture choices, such as batch normalization or annealing learning rates, prevented by the current setting.

**Choice of hyperparameters.** Our study exhibits a limited range of batch sizes and learning rates, along with unconventional grid values that varied between different architectures. These choices were made to align with, and ensure a fair comparison with, the experiments of [Dupuis et al., 2023]. We believe that these design choices were made by Dupuis et al. [2023] to ensure convergence of the models within reasonable numbers of iterations, due to the computational cost of repeatedly training different models with various seeds, and may have contributed to the statistically significant results reported in their work. For an extended analysis, we would explore a wider range of hyperparameters.

**Computational limitations.** Most of the runtime in our experiments was devoted to computing the loss-based PH dimension. Though efficient computation of PH is an active field of research in TDA Chen and Kerber [2011], Bauer et al. [2014a,b], Guillou et al. [2023], Bauer [2021], PH remains a computationally intensive methodology, limiting the number of experiments it was possible to run.

**Lack of identifiable patterns in the correlation failure.** An important limitation of our work is our failure to identify any clear patterns offering explanations as to when and why the PH dimension can fail to correlate with the generalization gap when conditioning on the network hyperparameters. We further cannot identify a pattern to explain the success and failure of PH dimension measures to correlate with generalization when using adversarial initialization.

**Theoretical limitations.** Despite providing extended experimental analyses of relationship between PH dimension and generalization gap, we do not make any theoretical contributions to explain the disconnect observed between theory and practice.

# 8   Conclusion

In this work, we extend previous evaluations of PH dimension-based generalization measures. Although theoretical results on the fractal and topological measures of generalization gaps were provided by Şimşekli et al. [2020], Birdal et al. [2021], Camuto et al. [2021] and Dupuis et al. [2023], experimentally, our study shows that there is in some cases a disparity between theory and practice. We suggest two directions for further investigation: (i) considering probabilistic definitions of fractal dimensions [Adams et al., 2020, Schweinhart, 2020] may offer a more natural interpretation for generalization compared to metric-based approaches; (ii) exploring multifractal models for optimization trajectories could better capture the complex interplay of network architecture and hyperparameters in understanding generalization. Overall, our work demonstrates that there is still much to understand concerning the complex interplay between generalization gap and TDA-based fractal dimension of optimization trajectories.

## Acknowledgments and Disclosure of Funding

The authors wish to thank Tolga Birdal, Justin Curry, Robert Green, and Sara Veneziale for helpful discussions. I.G.R. is funded by a London School of Geometry and Number Theory–Imperial College London PhD studentship, which is supported by the EPSRC grant No. EP/S021590/1. Q.W. is funded by a CRUK–Imperial College London Convergence Science PhD studentship, which is supported by Cancer Research UK under grant reference CANTAC721\10021 (PIs Monod/Williams). M.M.B. and C.B.T. are partially supported by EPSRC Turing AI World-Leading Research Fellowship No. EP/X040062/1. M.M.B. and A.M. are supported by the EPSRC AI Hub on Mathematical Foundations of Intelligence: An "Erlangen Programme" for AI No. EP/Y028872/1.

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

# A  Persistent Homology and Vietoris–Rips Filtrations

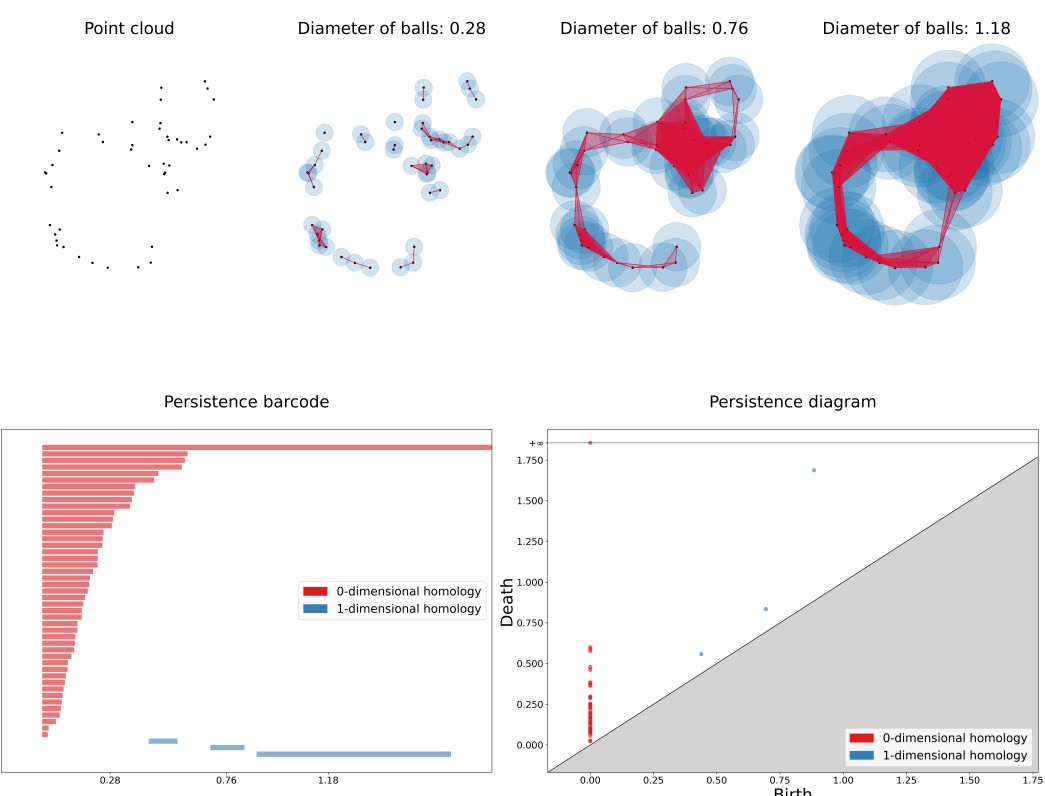

Figure 5: Vietoris–Rips filtration over two noisy circles (with 30 and 15 points each) at 4 different filtration values; and corresponding persistence barcode and diagram (0-dimensional PH in red, 1-dimensional PH in blue). Images produced using GUDHI [The GUDHI Project, 2020].

PH is a methodology for computing topological representations of data. It achieves this using a *filtration*, producing a compact topological summary of the topological features over this filtration often presented in the form of a *persistence barcode* or *persistence diagram*. Here, we briefly overview these key concepts. For a more complete introduction to PH see Zomorodian and Carlsson [2005], Oudot [2017].

A *simplicial complex* is a combinatorial object built over a finite set and defined as a family of subsets of such finite set, called simplices, which is closed under inclusion. Geometrically, this can be understood as a set of vertices, edges, triangles, tetrahedra, and higher-order geometric objects— i.e. higher dimensional simplices. Being closed under inclusion means, for instance, that if a triangle is present in the family, all the edges and vertices in the boundary of the triangle also belong to the complex. For finite subset $S \subset X$ of a metric space $(X, \rho)$, an example of such an object is the Vietoris–Rips [Vietoris, 1927] simplicial complex at scale $t \in [0, +\infty)$, defined as the family of all simplices of diameter less or equal than $t$ that can be formed with the finite set $S$ as set of vertices.

A *filtration* is defined as a family of nested simplicial complexes, that is, a parameterized set $\{K_t : t \in T\}$ with totally ordered indexing set $T$, such that if $s \leq t$ then $K_s \subset K_t$. The *Vietoris– Rips filtration* is then the family of Vietoris–Rips complexes at all scales $t \in [0, +\infty)$. More generally, a filtration is a family $\{F_t : t \in \mathbb{R}\}$ of nested simplicial complexes indexed by the real numbers, that is, if $t \leq s$ then we have $F_t \subset F_s$.

The *persistence barcode* provides a compact summary of the lifetime of topological features (components, holes, voids and higher dimensional generalizations) as we allow the filtration parameter to evolve. It is defined precisely as a multiset of bars, each of them spanning the lifetime of a topological

feature of the corresponding dimension. An alternative representation of the barcode is given by the *persistence diagram*. This is a scatterplot of points in the first quadrant of the two-dimensional plane, in the region above the diagonal, where each point is in correspondence with a bar in the barcode and has as the first coordinate its starting point and as the second coordinate the ending point of the bar. An example of a Vietoris–Rips filtration and the corresponding persistence barcode and diagram can be found in Figure 5. The Vietoris–Rips filtration is often utilized due to its fast computation [Bauer, 2021] and is also the method of choice for Dupuis et al. [2023] in computing the PH dimensions.

## B   Fractal Dimensions

Fractal dimensions describe the geometry of fractals—spaces that are rough and irregular on a finer scale. Informally, we want to say that such an object has dimension $d$ when its "local geometry" at scale $\epsilon$ scales as $\epsilon^d$ or $\epsilon^{-d}$, for some positive real number $d$ which need not be integral. Fractal shapes arise in a number of situations, such as spaces built from self-similar recursions, chaotic dynamical systems, and even real data.

We now review several notions of fractal dimension that are interesting for the purposes of this work, following the presentation in Adams et al. [2020], where many of the original references can be found. We begin by providing the definitions of *metric* fractal dimensions, defined in terms of subsets $S$ of a metric space $(X, \rho)$. The first fractal dimension of this kind to appear in the literature was the following.

**Definition B.1.** Let $d \in [0, \infty)$. The *d-Hausdorff measure* of $S$ is

$$H_d(S) := \inf_{\delta > 0} \left( \inf \left\{ \sum_{j=1}^{\infty} \mathrm{diam}(B_j)^d : S \subseteq \bigcup_{j=1}^{\infty} B_j \text{ and } \mathrm{diam}(B_j) \leq \delta \right\} \right)$$

where the inner infimum is taken over all coverings of $S$ by balls $B_j$ of diameter at most $\delta$.

**Definition B.2.** The *Hausdorff dimension* of $S$ is

$$\dim_{\mathrm{H}}(S) := \inf_d \{ H_d(S) = 0 \}. \tag{4}$$

In practice, it is difficult to compute the Hausdorff dimension, which lead to the introduction of more computable notions of fractal dimension. Let $N_\epsilon$ be the infimum over the number of balls of radius $\epsilon > 0$ required to cover $S$.

**Definition B.3.** The *box-counting dimension* of $S$ is

$$\dim_{\mathrm{box}}(S) = \lim_{\epsilon \to 0} \frac{\log(N_\epsilon)}{\log(1/\epsilon)} \tag{5}$$

provided this limit exists. Replacing the limit with a lim sup yields the *upper* box-counting dimension, and a lim inf gives the *lower* box-counting dimension.

There is also a notion of metric fractal dimension defined in terms of minimal spanning trees. Let $T(\mathbf{x})$ denote the minimal spanning tree of a finite subset $\mathbf{x} := \{x_1, \ldots, x_n\} \subset X$ and define

$$E_\alpha^0(\mathbf{x}) = \frac{1}{2} \sum_{e \in T(\mathbf{x})} |e|^\alpha.$$

**Definition B.4.** Let $S$ be a bounded subset of a metric space $(X, \rho)$. The *minimal spanning tree dimension* of $S$ is

$$\dim_{\mathrm{MST}}(S) := \inf \left\{ \alpha : \exists\, C > 0 \text{ s.t. } E_\alpha^0(\mathbf{x}) < C,\, \forall\, \mathbf{x} \subset S \text{ finite subset } \right\} \tag{6}$$

Many authors [Adams et al., 2020, Schweinhart, 2021, 2020] extended these ideas using persistent homology. We first present the approach followed in Schweinhart [2021]. Let $\mathbf{x} = \{x_1, \ldots, x_n\}$ be a finite metric space. Call $\mathrm{PH}_i(\mathbf{x})$ the persistence diagram obtained from the PH of dimension $i$ computed from the Čech complex of $\mathbf{x}$ and $\widetilde{\mathrm{PH}}_i(\mathbf{x})$ the one obtained from the Vietoris–Rips filtration. For any of these persistence diagrams we can define

$$E_\alpha^i(\mathbf{x}) := \sum_{(b,d) \in \mathrm{PH}_i(\mathbf{x})} (d - b)^\alpha. \tag{7}$$

**Definition B.5.** Let $S$ be a bounded subset of a metric space $(X, \rho)$. The *ith persistent homology dimension* ($\mathrm{PH}_i$-dim) for the Čech complex of $S$ is

$$\dim_{\mathrm{PH}}{}^i(S) := \inf \left\{ \alpha : \exists\, C > 0 \text{ s.t. } E_\alpha^i(\mathbf{x}) < C, \forall\, \mathbf{x} \subset S \text{ finite subset} \right\}. \tag{8}$$

It is possible to analogously define $\dim_{\widetilde{\mathrm{PH}}}{}^i(S)$ for the Vietoris–Rips PH.

In addition to these metric notions, we also have *probabilistic* fractal dimensions, defined directly in terms of a measure $\mu$ supported in a subspace $S$ of a metric space $(X, \rho)$.

**Definition B.6.** The *lower Hausdorff dimension* of a measure $\mu$ with total mass 1 is

$$\underline{\dim_{\mathrm{H}}}(\mu) = \inf\{\dim_{\mathrm{H}}(S) : S \text{ is a Borel subset with } \mu(S) > 0\} \tag{9}$$

while the *upper Hausdorff dimension* is

$$\overline{\dim_{\mathrm{H}}}(\mu) = \inf\{\dim_{\mathrm{H}}(S) : S \text{ is a Borel subset with } \mu(S) = 1\}. \tag{10}$$

**Remark 1.** *We have $\underline{\dim_{\mathrm{H}}}(\mu) \leq \dim_{\mathrm{H}}(\mathrm{supp}(\mu))$ and this inequality can be strict.*

A probability measure $\mu$ defined in a metric space $(X, \rho)$ induces a probability measure $\nu$ on the distance set of $X$, $\mathrm{dist}_X := \{\mathrm{dist}(x, y) : x, y \in X, x \neq y\}$

**Definition B.7.** The *correlation integral* of X is defined as the cumulative density function of $\nu$

$$C(\epsilon) := \mathbb{P}_\nu \left( \rho(x, y) \leq \epsilon \right).$$

The *correlation dimension* is thus defined as the limit

$$\dim_{\mathrm{corr}}(\mu) = \lim_{\epsilon \to 0} \frac{\log(C(\epsilon))}{\log(\epsilon)} \tag{11}$$

As a final remark, the PH dimension can also be seen as a probabilistic fractal dimension if we consider that the finite metric spaces $\mathbf{x}$ used Definition B.5 are coming from i.i.d. samples drawn from a probability measure $\mu$ supported on a subset $S$ of a metric space $(X, \rho)$. This is the approach followed in Schweinhart [2020], Jaquette and Schweinhart [2020], which introduce the following modified definition.

**Definition B.8** ([Schweinhart, 2020]). The *persistent homology dimension* ($\mathrm{PH}_i$-dimension) of a measure $\mu$, for each $\alpha > 0$ and $i \in \mathbb{N}$, is

$$\dim_{\mathrm{PH}}{}^{i,\alpha}(\mu) := \frac{\alpha}{1 - \beta} \tag{12}$$

where

$$\beta = \limsup_{n \to \infty} \frac{\log(\mathbb{E}(E_\alpha^i(\mathbf{x})))}{\log(n)}$$

where $\mathbf{x} = \{x_1, \ldots, x_n\}$ is set of i.i.d. samples drawn from $\mu$.

## B.1 Relations Between Different Notions of Fractal Dimension

It is worth mentioning here that a notion of fractal dimension does not need to be well-defined for every subset of a metric space or measure supported on it. In fact, there are shapes that present a "multifractal" structure scaling at several values of $d$ in our informal intuition above. However, if we assume the following regularity condition, the Hausdorff, box-counting, and correlation dimension are well-defined and all coincide.

**Definition B.9.** A probability measure $\mu$ supported on a metric space $(X, \rho)$ is *d-Ahlfors regular* if there exist $c, \delta_0 \in \mathbb{R}_+$ such that

$$\frac{1}{c}\delta^d \leq \mu(B_\delta(x)) \leq c\delta^d$$

for all $x \in X$ and $d < \delta_0$, where $B_\delta(x) := \{y \in X : \rho(x, y) < \delta\}$.

If $\mu$ is $d$-Ahlfors regular on $X$ then it is comparable to the $d$-dimensional Hausdorff measure in $X$ and the Hausdorff measure is also $d$-regular.

On the other hand, Kozma et al. [2006] prove that for any metric space (with no mention to any regularity conditions) the minimal spanning tree equals the upper box dimension

$$\dim_{\mathrm{box}}(S) = \dim_{\mathrm{MST}}(S). \tag{13}$$

Concerning minimal spanning trees and PH, there is a bijection between the edges of the Euclidean minimal spanning tree of a finite metric space $\mathbf{x} = \{x_1, \ldots, x_n\}$ and the intervals in the PH of $\mathrm{PH}_0(\mathbf{x})$, where we need to halve the length of the intervals in the PH decomposition. For the Vietoris–Rips PH $\widetilde{\mathrm{PH}}_i$ there is no need to halve the length of the intervals. In any case, it is clear from the definitions that

$$\dim_{\mathrm{PH}}{}^0(S) = \dim_{\widetilde{\mathrm{PH}}}{}^0(S) = \dim_{\mathrm{MST}}(S). \tag{14}$$

For higher homological degree, Schweinhart [2021] obtains conditions under which the PH dimension of higher homological degrees agrees with the box-counting dimension, in a similar vein to Kozma et al. [2006].

On the other hand, Schweinhart [2020] additionally presented a thorough study of the asymptotic behavior of the quantities $E_\alpha^i(\mathbf{x})$ for $i \in \mathbb{N}$, $\alpha > 0$, and $\mathbf{x} = \{x_1, \ldots, x_n\}$ a set of random i.i.d. samples drawn from a probability measure $\mu$ defined a metric space $(X, \rho)$. For instance, the following result concerning minimal spanning trees is proved.

**Theorem B.1** (Theorem 3, [Schweinhart, 2020]). Let $\mu$ be a $d$-Ahlfors regular measure on a metric space and $\mathbf{x} = \{x_1, \ldots, x_n\}$ i.i.d. samples from $\mu$. If $0 < \alpha < d$, then

$$E_\alpha^0(\mathbf{x}) \approx n^{\frac{d-\alpha}{d}}$$

with high probability as $n \to \infty$, where $\approx$ means that the ratio of the two quantities is bounded between positive constants that do not depend on $n$.

The hypothesis of $d$-Ahlfors regularity is not strictly needed in the proof of this theorem: it is possible to just assume weaker statements on the measure $\mu$ to prove the bounds. However, it is argued that $d$-Ahlfors regularity is included in the theorem because in an accompanying paper [Jaquette and Schweinhart, 2020], where the authors explore these quantities in applications, they observe that for fractals emerging from chaotic attractors (that do not satisfy Ahlfors regularity), for each $\alpha > 0$ there is a different value of $d_\alpha$ such that $E_\alpha^0(\mathbf{x}) \approx n^{\frac{d_\alpha - \alpha}{d_\alpha}}$. In particular, this means that we cannot replace $d$ in the theorem above with the upper-box or Hausdorff dimension.

Other results regarding the asymptotic behavior for $E_\alpha^i(x_1, \ldots, x_n)$ are also derived in Schweinhart [2020], where $d$-Ahlfors regularity is also required, in addition to some conditions on the asymptotic behavior of the expectation and variance of the $\mathrm{PH}_i$-dimension. Finally, Schweinhart [2020] establishes a correspondence between the $\mathrm{PH}_0$-dimension and the Hausdorff dimension when a measure is $d$-Ahlfors regular for $\dim_{\mathrm{PH}}{}^{0,\alpha}$, if $0 < \alpha \leq d$. There is also a connection for higher-order $\mathrm{PH}_i$-dimensions adding some extra conditions: requiring the measure to be defined in an Euclidean space and some asymptotic behavior for the expectation and variance of $\mathrm{PH}_i(x_1, \ldots, x_n)$.

## C  Additional Experimental Details

In this appendix we include additional experimental specifications for the experiments.

### C.1  Architectures

- As in Dupuis et al. [2023], both FCN-5 and FCN-7 networks have a width of 200 for each hidden layer and use ReLU activation.
- AlexNet follows the construction outlined in Krizhevsky et al. [2017].
- The standard CNN defined in consists of four $3 \times 3$ convolutional layers with widths $[c, 2c, 4c, 8c]$, where $c$ is a width (channel) multiplier [Nakkiran et al., 2021]. In all experiments, $c = 64$ unless otherwise stated. Each convolution is followed by a ReLU activation and a MaxPool operation with kernel = stride = $[1, 2, 2, 8]$.

## C.2 Training Configuration

We train using mean squared error for the regression experiments and cross-entropy loss for the classification experiments. Similarly to Dupuis et al. [2023] we report accuracy gap instead of loss gap for the classification experiments.

The convergence criteria follow Dupuis et al. [2023], and are as follows:

- For regression, we compute the empirical risk on the full training dataset every 2000 iterations and define convergence when the relative difference between two consecutive evaluations becomes smaller than $0.5\%$.
- For classification, we define convergence when the model reaches 100% training accuracy, given that the model is evaluated on the full training dataset every 10,000 iterations.

## C.3 Computation of PH Dimensions

The computation of the PH dimensions is based on Algorithm 1 by Birdal et al. [2021], using the code of Dupuis et al. [2023]. This codebase relies on the TDA software Giotto-TDA [Tauzin et al., 2021] to compute the PH barcodes in the two (pseudo-)metric spaces under study.

## C.4 Grid Experiments

All experiments for the correlation analysis utilize learning rates and batch sizes on a $6 \times 6$ grid, defined by Dupuis et al. [2023] and repeated below:

- For CHD, learning rates are logarithmically spaced between 0.001 and 0.01. Batch sizes take values $\{32, 65, 99, 132, 166, 200\}$.
- For MNIST and CIFAR-10, learning rates are logarithmically spaced between 0.005 and 0.1. Batch sizes take values $\{32, 76, 121, 166, 211, 256\}$.

Experiments on MNIST and CHD are repeated with seeds $\{0, \dots, 9\}$, while experiments on CIFAR-10 use seed 0.

## C.5 Adversarial Initialization

A batch size of 128 is used for all experiments, with a constant learning rate of 0.01. We train using seeds $\{0, \dots, 29\}$ for MNIST and AlexNet CIFAR-10, only seeds $\{0, \dots, 19\}$ are used for CNN CIFAR-10 due to computational constraints.

## C.6 Model-wise Double Descent

We train the standard CNN with CIFAR-100 with a constant learning rate of 0.01, a batch size of 128, and seeds $\{0, 1, 2\}$. Since not all model widths achieve 100% training accuracy on CIFAR-100, we terminate training after 250,000 iterations.

# D Stability of PH Dimension Estimates

The theory proposed by Birdal et al. [2021] and Dupuis et al. [2023] establishes PH dimensions as a measure of generalization, but in practice we can only compute an approximation of this quantity. As detailed in Section 3, we compute this estimation using the 5,000 iterations after reaching convergence $\{w_k : 0 < k \leq 5000\}$. To study if this value remains constant after the first 5,000 iterations, we conduct an additional experiment in which we train for 15,000 iterations beyond the convergence criterion $\{w_k : 0 < k \leq 15000\}$. We then compute 3 estimations of the PH dimensions: for the first 5,000 iterations $\dim_{\mathrm{PH}}(w_{0:5000})$, using the samples between 5,000 and 10,000 iterations $\dim_{\mathrm{PH}}(w_{5000:10000})$ and between 10,000 and 15,000 iterations $\dim_{\mathrm{PH}}(w_{10000:15000})$. We then compute rank correlation coefficients between the three estimates, to evaluate their relative order, and their relative difference by computing the following quantity

$$\frac{\dim_{\mathrm{PH}}(w_{j:k}) - \dim_{\mathrm{PH}}(w_{p:q})}{\dim_{\mathrm{PH}}(w_{j:k})} \cdot 100.$$

Table 4 contains the results obtained. We note that the relative differences between values are small, and do not have a consistent sign, indicating a (small) fluctuating pattern that is likely due to randomness. Concerning the rank correlations, they seem to indicate that the relative ordering is consistent between different sets of iterations.

Table 4: Relative difference (%) and Spearman's $\rho$ and Kendall $\tau$ rank correlations for the PH dimensions estimates computed using different subsets of trajectory after convergence. Means over 10 seeds of each model are presented with standard deviations as error bars.

| Model & Data | Measures | 5,000 vs 10,000 | | | 10,000 vs 15,000 | | | 5,000 vs 15,000 | | |
|---|---|---|---|---|---|---|---|---|---|---|
| | | % difference | $\rho$ | $\tau$ | % difference | $\rho$ | $\tau$ | % difference | $\rho$ | $\tau$ |
| FCN-5 CHD | Euclidean | $0.15 \pm 1.74$ | 0.63 | 0.43 | $-0.41 \pm 1.05$ | 0.88 | 0.72 | $-0.25 \pm 1.72$ | 0.68 | 0.53 |
| | Loss based | $-3.83 \pm 6.82$ | 0.75 | 0.56 | $-3.48 \pm 6.62$ | 0.62 | 0.49 | $-7.24 \pm 7.44$ | 0.72 | 0.55 |
| FCN-5 MNIST | Euclidean | $0.11 \pm 0.66$ | 0.65 | 0.46 | $0.18 \pm 0.60$ | 0.46 | 0.34 | $-0.07 \pm 0.74$ | 0.45 | 0.33 |
| | Loss based | $1.53 \pm 1.35$ | 0.62 | 0.45 | $1.01 \pm 1.21$ | 0.53 | 0.38 | $2.62 \pm 1.22$ | 0.65 | 0.48 |
| AlexNet CIFAR-10 | Euclidean | $-0.75 \pm 0.63$ | 0.86 | 0.68 | $0.02 \pm 0.40$ | 0.87 | 0.69 | $-0.72 \pm 0.64$ | 0.92 | 0.78 |
| | Loss based | $-1.05 \pm 2.19$ | 0.61 | 0.46 | $-1.41 \pm 1.67$ | 0.23 | 0.16 | $-2.47 \pm 2.77$ | 0.14 | 0.11 |

# E   Additional Experimental Results

In Figure 6, we present the results for the FCN-5 used for the computation of the corresponding correlation coefficients in Table 1, that were not present in Figure 2 in the main text due to space constraints. In Figure 7 we additionally plot $||w_{5000}||_2$ against the generalization gap for the results of Table 1.

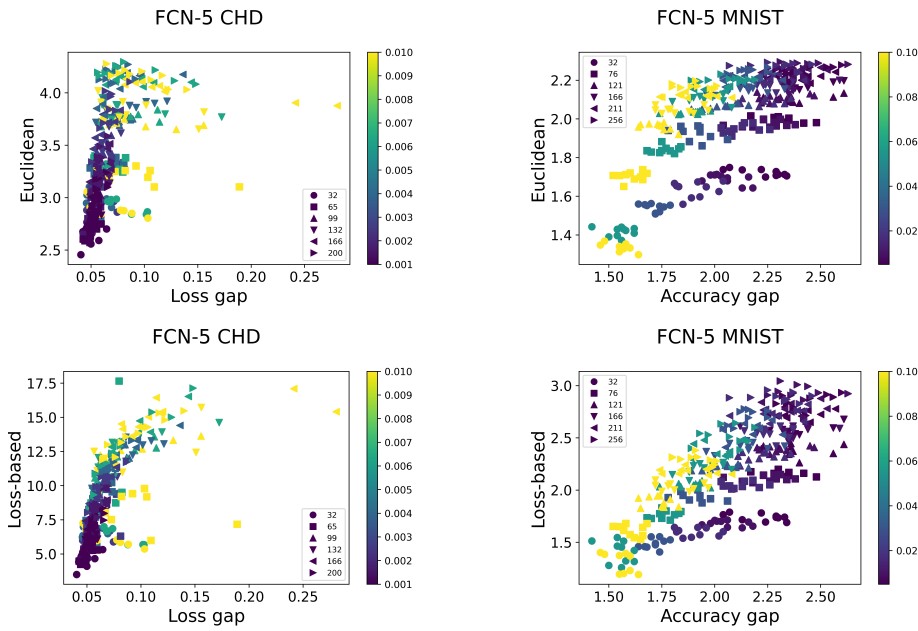

Figure 6: **Learning rate/batch size grid results for FCN-5 architecture.** Euclidean (top) and loss-based (bottom) PH dimension plotted against generalization gap for range of learning rates and batch sizes for FCN-5 architecture.

# F   Conditional Mutual Information

The CMI, denoted by $I$, for discrete random variables is defined as

$$I(X;Y|Z) = \sum_{z \in \mathcal{Z}} p_Z(z) \sum_{y \in \mathcal{Y}} \sum_{x \in \mathcal{X}} p_{X,Y|Z}(x,y|z) \log \frac{p_{X,Y|Z}(x,y|z)}{p_{X,Z}(x,z)p_{Y,Z}(y,z)},$$

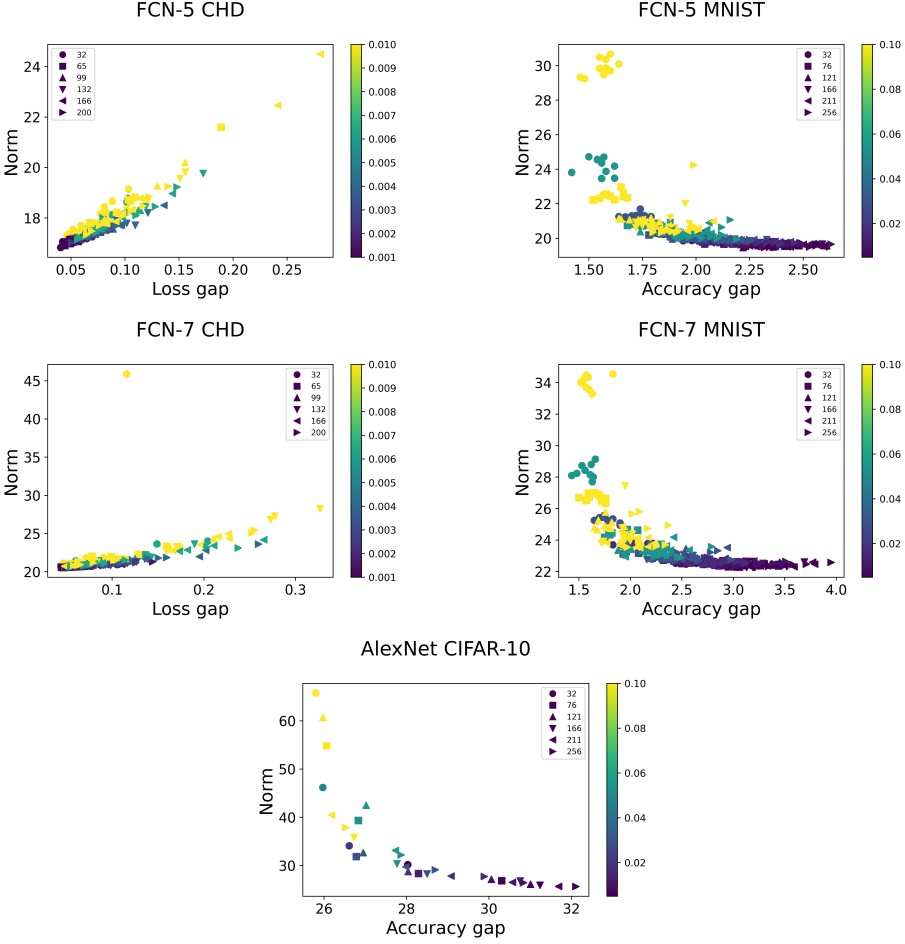

Figure 7: **Learning rate/batch size grid results for parameter norm.** $\|w_{5000}\|_2$ plotted against generalization gap for range of learning rates and batch sizes.

where $p$ is the empirical measure of probability. From this definition, we see that the CMI vanishes if and only if $X \perp Y \mid Z$, i.e., $X$ and $Y$ are conditionally independent given $Z$. Hence, while changes in $X$ might seem linked to changes in $Y$, the CMI allows us to isolate the effect of $Z$ and establish whether $X$ and $Y$ are independent when $Z$ is fixed. In Section 4.3, $X$ is the PH dimension, $Y$ the generalization error, and $Z$ the learning rate.

