# OpenReview forum: "On the Limitations of Fractal Dimension as a Measure of Generalization"
_NeurIPS.cc/2024/Conference — NeurIPS 2024 poster_

### Official Review · Reviewer_d6bB · 2024-07-04

**Soundness:** 4
**Presentation:** 2
**Contribution:** 3
**Rating:** 7
**Confidence:** 4

**Summary:**

This study revisits recent claims about persistent homology dimension (in the limit a fractal dimension) and the generalization gap of supervised learning systems. The essential theoretical background is summarized, and then three main experiments are conducted, concerning 1) (partial) correlation of the concepts, 2) influence of poor or adversarial initialization, 3) double-descent phenomenon. Overall, the results indicate that PH only weakly correlates with the generalization gap, and that hyper-parameters may be a confounder since there appears to be a stronger correlation between PH and hyper-parameters.

**Strengths:**

S1 - The considered topic is relevant to ML theory and the experiments were conducted with high quality. All formal statements are correct.

S2 - The reported results, and the discussion thereof, are insightful and useful to the wider community.

S3  - The discussion of limitations is wide and transparent. Section 5 is a role-model of how a paper's limitations should be presented.

**Weaknesses:**

W1- Readability. The work is not self-contained and heavily depends on information from other papers. This makes it tedious to study this work and requires the reader to frequently switch back and forth between multiple papers. Although several passages are well-written, the paper is difficult to follow overall. This may be, in part, because the paper is not self-contained, but the frequent cross-references within this paper suggest that the structure may not be optimal.

W2 - The hypothesis testing setup is not properly described. It is not true that the hyper-parameters must have a significant influence if we reject the null-hypothesis that there is a correlation between PH and the generalization gap. Lines 228-230 need to be re-written (and extended) so that the hypothesis test setup is clear and rigorous to avoid misinterpretations.

W3 - The title and abstract are misleading need to be updated to reflect on this work's experimental focus. I was expecting a theoretical paper. For example, the phrase "This work performs an extended evaluation of these topological generalization measures" could also be understood as a formal evaluation.

**Questions:**

Q1 - The considered hypothesis test of PH <-> generalization vs. PH <-> hyper-parameters actually feels like two hypothesis tests, where each considers the absence of the respective correlation separately. Please explicitly list the considered hypotheses and consider if such as split makes sense.

Q2 - Please consider the improvement suggestions below.

**Improvement suggestions**

line 65: This is a slight abuse of notation. I would prefer if the composed loss function were explicitly declared as such, i.e. $\ell(w,z)=\mathcal{L}(h_w(x),y)$. Otherwise, the paper is not self-contained.

line 80: It is stated that further details on persistent homology are given in the supplementary material, but actually it is in the appendix. This is good for a camera-ready version, but in the review phase it is a bit confusing.

line 82: **x** is not a metric space, it is a set. A metric space is a tuple of a set and a metric. Please state what the metric is – presumably $\rho$?.

typos in lines 89 & 106

Eq. (2): $w'$ is not defined here. Also, maybe use a different notation, since $\rho$ is later used for Spearman correlation.

lines 144-154: I would appreciate if at least some of the selections were explained, e.g., why cross-entropy loss?

line 172: This needs to be explained further. I understand that long trainings needs to stop at some point, but an interruption may influence the results if done poorly.

line 233: "causally correlated" -> "correlated".

lines 244-245 can be omitted since they have the same content as 242-243.

In Table 2, it might be helpful to use colors or bold fonts to visually distinguish between significant and non-significant findings, i.e., p <= 0.05 or not.

Definition of mutual information: Maybe use a different notation for probability like $\text{Pr}(\cdot)$ , since $p$ is already the $p$-value.

line 276: Grammar mistake, "PH dimensions ... does"

Section 4.3. This discussion is feels too short. Maybe condense lines 213-243 to make more space.

line 290: Typo? Please rephrase.

line 291: typo "preventted" -> "prevented"

**Limitations:**

The limitations are properly addressed

---

> ### Author Rebuttal · Authors · 2024-08-06
>
> We thank this referee for their careful, detailed feedback. We are pleased to see that they found our experiments “conducted with high quality”, the results “insightful”, “useful” and “relevant” to the community, and that they appreciated our discussion on limitations as a “role-model” on how these should be discussed. We now address the weaknesses identified.
>
> **W1. Readability.** We acknowledge that our study involves a certain amount of previous background, which is deep and appears across several papers. We have done our best to summarize it in the clearest way possible given the page restriction. We understand that thismay be insufficient for the unfamiliar reader, we will improve the readability and accessibility by including an extended version of the background in the supplementary material, which aims to be self-contained and to serve as an introduction to this line of work. We will also revise the structure of the paper to minimize the number of cross-references, and to streamline some of the discussions.
>
> **W2. Hypothesis in permutation tests**: We thank the reviewer for pointing this out, indeed there was a typo and insufficies in the way we had described the hypothesis. We would like to rewrite the beginning of this section to:
> Line 221: *\[...\] we conduct a non-parametric permutation-type hypothesis test for the assumption that the partial correlation, i.e., the correlation coefficient between the regression residuals, is equal to zero. Transferred to our setting, the null hypothesis implies that the correlation observed between generalization and PH dimensions is explained by the influence of other hyperparameters.*
> Lines 228-230, where we explain how to interpret the p-values from the permutation test, will be rewritten to:
> *Recall that a p-value lower than 0.05 implies the rejection of the null hypothesis, or equivalently, it implies that there is a correlation between PH dimension and generalization gap that cannot be explained by the influence of the hyperparameters; whereas a p-value bigger than 0.05 implies a significant influence of the corresponding hyperparameter in the apparent correlation.*
> We have also included an interpretation of these results, motivated by the answer to reviewer garn, which can be consulted in the corresponding answer.
>
> **W3. Update abstract**: We acknowledge that the abstract might have created an expectation for work of a different nature, so we are open to change it.
>
> We now address the question posed by the reviewer.
>
> **Q1.** We thank the reviewer for this question. We implement two hypothesis tests in our work: one to report the statistical significance of the partial correlation test and the other for the conditional independence test.
> - The null hypothesis of the former is that there is an influence of the hyperparameters in the observed correlation between generalization and PH dimensions. The null hypothesis is specified above. In this case we are not considering the correlation between generalization gap and PH dimensions directly, we are rather trying to evaluate the effect of the learning rate on it.
> - Concerning the conditional independence test, the null and alternative hypotheses are illustrated in the Appendices, Section F, Figures 3 and 4. Here the null hypothesis is conditional independence between PH dimensions and generalization gap conditioned on the hyperparameters of the network. We could move the images to the main text if this makes it more clear.
>
> **Q2.** We now cover all the suggestions for improvement.
> - We have fixed all typos and redundant information, and would like to reiterate our thanks to the reviewer for this very careful read of our work, which was really helpful in improving it.
> - We agree with the suggestion about the composed loss function, and will change it in an updated version of the paper.
> - We notice that we might have confused appendices and supplementary material; we will refer to the text coming after the references as appendices instead of supplementary material.
> - We thank the reviewer for this observation, indeed, we forgot to specify the metric. We would like to change the sentence to: *Let $\\mathbf{x} \= \\{x\_1, \\, \\dots, \\, x\_n\\}$ be a finite subset of a metric space $(X, \\rho)$.*
> - We will include the definition of $w’$ in Eq. (2) and will think of a suitable change of notation for the metric.
> - The justification for our architectural choices in the experimental section is mainly that we are closely imitating the configuration of the experiments in Dupuis et al. \[2023\] and Birdal et al. \[2021\], to make the comparison fair, and we tried to justify them in the Limitations section, within Models and Settings. This involves the networks, batch sizes, learning rates and datasets in the correlation analysis. The cross-entropy loss is used because we need it to be continuous in both variables (the parameter and the evaluation point) and uniformly bounded (see Assumption 3.1 in Dupuis et al. \[2023\]). The other architectural choice is the “standard cnn” that is designed to showcase double descent, which we implemented because we wanted to test if this affected the PH dimension.
> - We thank the reviewer for this observation, as we have realized that this sentence is a bit misleading, and will rewrite it in next iterations of the paper. We have the same stopping criterion than Dupuis et al \[2023\]: for classification, achieving 100% train accuracy; for regression, having the relative training loss to be smaller than 0.5% within 2000 iterations.
> - We have completely changed the line 233, we included the new one in a reply above.
> - We thank the reviewer for the suggestion of bolding relevant results in tables, this is already reflected in the revised table 1 of the global response and will be implemented for all other tables.
> - We thank the reviewer for the observation on the notation for probabilities in the CMI, we will change it to avoid confusion.

---

> > ### Comment · Reviewer_d6bB · 2024-08-08
> > **Acknowledgement of author rebuttal**
> >
> > I thank the authors for their rebuttal. Overall, I think they did a good job addressing the points I raised. I am also happy with the overall response, and how the authors handled the points raised by the other reviewers.
> >
> > The proposed changes for W2 seem sufficient. I agree with the authors that W1 is something that probably can't be easily fixed.
> >
> > Regarding the abstract, it might be sufficient to rewrite the fourth sentence as *This work performs an ~extended~ **empirical** evaluation of these topological generalization measures*.
> >
> > I will keep my positive rating.

---

> > > ### Author Response · Authors · 2024-08-09
> > > **Reply to Acknowledgement of author rebuttal**
> > >
> > > We are very grateful to the reviewer for their reply and are happy to see that they found our answers and the changes that we proposed satisfactory. We will implement these alongside the modification in the abstract suggested by the reviewer in a revised version of the paper.
> > >
> > > We are very pleased that the reviewer remains positive and maintains their acceptance rating.  Thank you again for all you support and the time devoted to helping us improve our work!

---

### Official Review · Reviewer_garn · 2024-07-10

**Soundness:** 4
**Presentation:** 3
**Contribution:** 3
**Rating:** 7
**Confidence:** 4

**Summary:**

This work empirically studies the connection between the generalization gap and the fractal dimension of a neural network's optimization trajectory. A recent line of research has established such connections in a theoretical manner through upper generalization bounds. Specifically, three types of experiments are performed
1. An analysis of the correlation between fractal dimensions and empirical generalization gap in a typical training scenario. Here, the experiments from prior work [1] are replicated and analysed to greater detail.
2. An analysis of such correlations for models that do not generalize properly, obtained from training starting at an adversarial initialization.
3. A double descent behavior of fractal dimensions is observed when training models of varying width.

In the first two experiments, only few statistical significance of the correlation between fractal dimensions and generalization is observed, i.e., less than between generalization and the L2 norm of parameters.

**Strengths:**

- The empirical evidence presented in prior work is closely examined and questioned. While this might not be as appreciated as the derivation of novel theoretical results or of previously unobserved empirical findings, it is definitely necessary in order to establish a solid foundation future work can build upon.
- There is substantial effort in identifying and transparently communicating the limitations of this work, much more than in all other submissions I have reviewed for NeurIPS 2024.
- Multiple different correlation scores are computed, and the findings are tested for statistical significance.
- The background section provides a good overview over the research on generalization bounds in terms of fractal dimensions.

**Weaknesses:**

- In section 4.1, the experiments from [1] are carefully replicated. However, the observations do not match the ones from [1]. For example, for FCN-5 on CHD, a *negative* correlation score of $\rho = -0.7622$ is reported, whereas [1] reports a *positive* correlation of $\rho = 0.87$, which looks plausible from [1, Figure 2]. This inconsistency is surprising, but it is neither discussed nor mentioned.
If the setting is indeed equal, then the correlation coefficients should be much more similar. I do trust both works in that they correctly report their results, but this contradiction needs to be resolved in some form.

- This submission presents the empirical data, but is very cautious in interpreting it. On the one hand, it is good to let the data speak for itself, but on the other hand, the relevant part is how the observations relate to the larger context.
  For example, at the end of section 4.1 (Correlation Analysis), it is stated that "Using this method, we conclude that the correlation coefficients between generalization and the norm or the ratio of learning rate over batch size are indeed significantly higher than the correlation coefficients with PH dimensions [l.209]"
  What are the conclusions from this? Similar questions apply to the other sections.

- It seems that the data is collected from only 10 random seeds (in each setting). For a purely empirical work that seems a bit low, especially if strong claims are deduced from it, e.g. "We demonstrate that fractal dimension fails to predict generalization of models trained from poor initializations"

- Not a weakness of the submission (rather on my end), but worth mentioning: I do not have sufficient knowledge on testing for correlations or independence. Thus, I will not comment on the used statistical pipeline (setup and choice of correlation coefficients, null hypotheses, tests, etc)

- typos:
  l.290 we were only work ...
  l.291 preventted

### References
1. Dupuis et al., Generalization Bounds using Data-Dependent Fractal Dimensions, ICML 2023.
2. Birdal et al., Intrinsic Dimension, Persistent Homology and Generalization in Neural Networks, NeurIPS 2021
3. Hofer et al., Connectivity-Optimized Representation Learning via Persistent Homology, ICML 2019
4. Chen et al., A Topological Regularizer for Classifiers via Persistent Homology, AISTATS 2019
5. Carrière et al., PersLay: a neural network layer for persistence diagrams and new graph
topological signature, AISTATS 2020

**Questions:**

- Your empirical observations contradict the theoretical predictions deduced from the generalization bounds in [1,2,..]. What do you think is the cause? Is it, because there is an error in the theory, are the assumptions in the theory unrealistic (not fulfilled in practice), is the fractal dimension not the decisive term in the generalization bounds, etc?

- A limitation that is not stated in your discussion section, but most likely plays a role, is the uncertainty of the fractal dimensionality estimates. I wonder, how much difference there is, if you continue SGD for another 5k iterations and re-estimate the dimension from them. Are the estimates approximately equal? Is their order the same? Is there any rank correlation between estimates from the first 5k and the second 5k iterations at all? Note that even if the estimates are equal, it does not mean that the estimates are accurate, as they might be biased.

- I guess, this was not your intention, but the remark "While efficient PH computation is a field in itself, integrating it with network training to make fractal dimension computations more feasible is an interesting direction, especially if we are able to derive a regularization term. [l312]" suggests that integrating persistent homology into neural network training hasn't been done before. This is not the case, and there are *many* works that do so, [3-5] to name a few. In fact, such a regularization term is even used by Birdal et al. [2] to encourage low ph dimensions.

- I did not understand your test of conditional independence, cf l.257. Would you explain it to me?

**Limitations:**

Yes, the authors adequately addressed the limitations!

---

> ### Author Rebuttal · Authors · 2024-08-06
>
> We thank this reviewer for the careful review provided. We are very happy to see that they also appreciated the importance of evaluating existing work to “build upon solid foundations”. We would like to begin by addressing the weaknesses they identify.
> 1. **Discrepancy between our correlation coefficients and those in Dupuis et al. \[2023\]**: We appreciate the reviewer's observation, which prompted us to revise the table. The updated table, along with detailed adjustments and answers, is included in the global response. Additionally, we have provided plots for the correlation experiment similar to those in Dupuis et al. \[2023\] to clarify the coefficients.
> 2. We acknowledge that our interpretations may have been conservative. Here is a summary, which will be included in the revised paper:
>    * **Extension of correlation analysis to other hyperparameter:**
>      1. Significant correlations with learning rate/batch size point out a potential confounding effect in the observed correlation between fractal dimension and generalization gap, which we closely examine in further tests.
>      2. The norm is a simple but flawed generalization quantifier and is significantly easier to compute than the PH dimensions. Observing higher correlations with the norm underscores the need to justify the use of more complex and less effective measures, which has not been addressed before.
>    * **Partial correlation:** there is a significant partial correlation conditioned on the learning rate, for fixed batch sizes, in most cases for the Euclidean PH dimension. This suggests that the observed correlation between generalization and PH dimension could be attributed to a correlation between PH dimension and learning rate. The trend for the loss-based dimension is less consistent, but it appears to be more robust to the partial correlation test.
>    * **Conditional independence test:** different results arise depending on the training dataset, pointing out that this might be another limitation when using PH dimensions as generalization measures.
>    * The **adversarial initializations** indicate that PH dimension fails to predict generalization. Models with poor generalization (higher generalization gap) due to adversarial training, show lower fractal dimensions. This directly contradicts the proposed theory that PH dimension should be lower for better generalizing models. A clear plot illustrating this effect is included in the global answer.
>    * Finally, the **double descent** manifesting with PH dimensions suggests some connection between the convergence properties of models in the critical region of double descent with respect to the model width, itself a poorly understood phenomenon, and the Euclidean persistent homology dimension. It also shows the failure of PH dimension to correlate with generalization gap, which stays monotonic in this region, while PH dimension not.
> 3. We agree the **number of seeds in the adversarial initialization** (10) was limited and have increased it to 30 seeds for AlexNet CIFAR10 and FC MNIST. Due to computational constraints we were only able to increase CNN CIFAR10 to 20 seeds. We refer to the global response for the results and interpretation in the corresponding figure. We hope this will help enhance the results.
>
> We now answer the questions posed.
> * We believe there are **two main reasons for our findings contradicting the generalization bounds**:
>    1. The assumptions required to prove generalization bounds and their implications on architecture choices and training data remain unclear. Different studies involve various technical assumptions about optimization trajectories and loss functions that may not be met in practice.
>    2. The primary reason for the discrepancy is likely the term in the generalization bound involving conditional mutual information between the training data and the optimization trajectory, which is complex and less understood compared to the fractal dimension term. It is unclear if this term can be empirically tested or estimated. The assumption that the conditional mutual information does not dominate the bound has not been proven and has been scarcely explored. We believe some experiments may enter a regime where this term dominates, disrupting the expected correlation.
> * **PH Dimension Estimations:** The reviewer's observation motivated us to implement suggested experiments. We refer to the global answer for further details.
> * We acknowledge that the line concerning the **integration of persistent homology into neural network training** is incorrect and due to an oversight in our writing, we will rectify it in the revised version of the paper. We also thank the reviewer for providing useful references that we will incorporate when expanding on this explanation.
> * We appreciate the **question about the CMI** and would like to elaborate on the brief outline previously provided.
>    1. From the definition on page 6, the CMI \\(I(X; Y \\vert Z)\\) vanishes if and only if \\(X \\perp Y \\vert Z\\), i.e. \\(X\\) and \\(Y\\) are conditionally independent given \\(Z\\). Thus, while changes in \\(X\\) might seem linked to changes in \\(Y\\), the CMI allows us to isolate the effect of \\(Z\\) and establish whether \\(X\\) and \\(Y\\) are independent when \\(Z\\) is fixed.
>    2. For our test, \\(X\\) is PH dimension, \\(Y\\) generalization error, and \\(Z\\) the neural network hyperparameters. Given the discrete nature of our selected hyperparameters, we empirically determine the probability density functions.
>    3. To assess the significance of the computed CMI, we generate a null distribution for the CMI under local permutations of \\(X\\) or \\(Y\\) for fixed hyperparameter values. The null hypothesis implies that \\(X\\) and \\(Y\\) are conditionally independent, so the CMI is invariant to permutations. We reject the assumption of conditional independence if the computed CMI lies in the extremes of the simulated null distribution.

---

> > ### Comment · Reviewer_garn · 2024-08-09
> >
> > I thank the authors for their rebuttal.
> >
> > The values in the updated Table 1 do not contradict the empirical results of Dupuis et al. [2023]: anymore. The previous **discrepancy to Dupuis et al. [2023]** was a major factor in my cautious rating and is now resolved. However, now the rank correlations to norms on CHD have a flipped sign. Would you comment on that?
> >
> > My second concern (limited **conclusions from the empirical findings**) has also been addressed satisfactorily. The sketched interpretations of results should certainly be added to the main part in an extended (but still nuanced) manner. As should be the two main conjectures of why the empirical findings seem to contradict the generalization bounds.
> >
> > Regarding the added **ph-dimension estimates**, my question has only partially been answered, in that rank correlations between ph-dimensions have been presented, which addresses the relative order of estimates. However, I was also asking whether the estimates are approximately equal at different numbers of SGD iterations, which concerns the actual values of the estimates. It also seems (to me) that comparing always to 5k iterations (i.e. 5k vs 10k and 5k vs 15k) instead of (15k vs 10k) would be more telling, as differences from 5k to 10k might propagate. Moreover, as the measured rank correlations are far from 1, I would be very interested in a variant of table 1, where 15k iterates are used. FCN-5 & CHD and FCN-5 &
> > MNIST would be sufficient.
> >
> > All **minor concerns** have been sufficiently addressed.
> >
> > Regarding points raised by the **other reviewers**; I agree with Reviewer d6bB that title and abstract are misleading should to be updated to reflect on the work's experimental focus.

---

> > > ### Author Response · Authors · 2024-08-12
> > > **Reply to Official Comment by Reviewer garn (1)**
> > >
> > > We are very grateful to the reviewer for their reply and are happy to see that they found our answers to most of their concerns satisfactory. We now address the two pending points that they mention in their message.
> > > - **Flip of sign of the correlation with norm**:
> > >     - Although there are challenges and limitations to the parameter norm as a measure of generalization [1], it is still a widely used and accepted measure of model complexity. We included this in our experiments as a simple, universally understood reference point for a high-complexity generalization measure. Future work could compare further complexity measures against these topological measures.
> > >     - Classical ML theory would suggest a positive correlation between the norm and generalization gap. On regression we observe this positive correlation, but on the classification we observe the inverse.
> > >     - It would appear that the norm correlates with the LB/ratio which itself is (weakly) positively correlated with gap on regression, and (strongly) negatively correlated with gap on classification. A contributing factor to the weak correlation between LB/ratio and gap on regression could be the limited range of learning rate values considered ($[10^{-3}, 10^{-2}]$).
> > >     - Finally, we note that the norm correctly correlates with generalization gap in all experiments using adversarial initialization and in the double descent experiment (not with evaluation accuracy as observed in the Euclidean dimension). We can present similar plots like the ones included in the general answer in the final version of the paper.
> > >     - We can also include relevant discussion when we present Table 1, we suggest the addition of the following paragraph: *Classical machine learning theory suggests a positive correlation between the norm and the generalization gap, which we observed in our regression experiments. However, in classification tasks, we found an inverse relationship, likely influenced by the learning rate (LR) and batch size (BS) ratio (LR/BS ratio), which may act as a confounding factor similar to effects seen with PH dimensions. Although the norm faces challenges and limitations as a generalization measure for deep networks [1], we included it in our comparisons as a simple, universally understood, and widely used and accepted reference point to compare with high-complexity generalization measures like the PH dimensions. We note that the norm correctly correlates with generalization gap in experiments with adversarial initialization and double descent, contrasting with PH dimensions (see Figures).*
> > > - **PH dimension estimates**: We apologize if the table presented was incomplete. We now present an updated version of the table, where:
> > >     - We include comparisons between 5k and 15k.
> > >     - We include a new column per case with the % difference, that is: $[(\text{PH dim at value 1}−\text{PH dim at value 2})/\text{PH dim at value 1}]∗100$. We have computed this difference per seed in each model, and present here the average and standard deviation as error bar. Looking at these new columns, one can note that the relative differences between values are small, and do not have a consistent sign, indicating a (small) fluctuating pattern that is likely due to randomness.
> > >
> > > |    Model & Data    |  Measures  |     5k vs 10k    |        |        |    10k vs 15k    |        |        |     5k vs 15k    |        |        |
> > > |:------------------:|:----------:|:----------------:|:------:|:------:|:----------------:|:------:|:------:|:----------------:|:------:|:------:|
> > > |                    |            | $\%$ difference  | $\rho$ | $\tau$ | $\%$ difference  | $\rho$ | $\tau$ | $\%$ difference  | $\rho$ | $\tau$ |
> > > |     FCN-5 & CHD    |  Euclidean | $0.15 \pm 1.74$  |  0.63  |  0.43  | $-0.41 \pm 1.05$ |  0.88  |  0.72  | $-0.25 \pm 1.72$ | 0.68   | 0.53   |
> > > |                    | Loss based | $-3.83 \pm 6.82$ |  0.75  |  0.56  | $-3.48 \pm 6.62$ |  0.62  |  0.49  | $-7.24 \pm 7.44$ | 0.72   | 0.55   |
> > > |    FCN-5 & MNIST   |  Euclidean | $0.11 \pm 0.66$  |  0.65  |  0.46  | $0.18 \pm 0.60$  |  0.46  |  0.34  | $-0.07 \pm 0.74$ | 0.45   | 0.33   |
> > > |                    | Loss based | $1.53 \pm 1.35$  |  0.62  |  0.45  | $1.01 \pm 1.21$  |  0.53  |  0.38  | $2.62\pm 1.22$   | 0.65   | 0.48   |
> > > | AlexNet + CIFAR-10 |  Euclidean | $-0.75 \pm 0.63$ |  0.86  |  0.68  | $0.02 \pm 0.40$  |  0.87  |  0.69  | $-0.72\pm 0.64$  | 0.92   | 0.78   |
> > > |                    | Loss based | $-1.05\pm 2.19$  |  0.61  |  0.46  | $-1.41 \pm 1.67$ |  0.23  |  0.16  | $-2.47 \pm 2.77$ | 0.14   | 0.11   |
> > >
> > > # References
> > >
> > > [1] Jiang, Y., Neyshabur, B., Mobahi, H., Krishnan, D., & Bengio, S. (2019). Fantastic generalization measures and where to find them. arXiv preprint arXiv:1912.02178.

---

> > > > ### Author Response · Authors · 2024-08-12
> > > > **Reply to Official Comment by Reviewer garn (2)**
> > > >
> > > > - **Table 1 using PH dimensions with 15k iterations**: In the table above we are computing PH dimension with 5k iterates, taking the weights of the 0-5k, the 5k-10k and the 10k-15k iterations after reaching 100% train accuracy. In response to the reviewer’s request for an updated version of Table 1 with PH dimension computed using all 15k iterations after reaching 100% training accuracy, we initiated these additional experiments. Unfortunately, we are limited by computational constraints during this discussion period and it will not be feasible to train and compute the dimension for a sufficient number of models by the end of the discussion period (with each model taking more than 1h to run, and a total of 720 models required for these new results).  We sincerely apologize for this.

---

> > > > ### Comment · Reviewer_garn · 2024-08-13
> > > >
> > > > Thank you again for your response.
> > > >
> > > > Regarding the **correlation with weight norms**. I was wondering, why the updated evaluation method for table 1 had a very strong effect on the correlation with weight norms (a positive sign in the correlation coefficient, compared to the negative sign in the original table 1 in the submission), but only on one dataset (CHD).
> > > > You make a good point however, that in theory, we would expect a *positive correlation* between norms and generalization gap, which is not seen in table 1, except on the CHD regression task. In this light, statements comparing weight norms and fractal dimensions as a measure of generalization need to be toned down, including (but not limited to) this sentence in the abstract: *"We further identify that the ℓ2 norm of the final parameter iterate, one of the simplest complexity measures in learning theory, correlates more strongly with the generalization gap than these notions of fractal dimension"*
> > > >
> > > > Regarding **fractal dimension estimates**. Thank you for the new table, please include this table in the appendix of the paper, as it shows (except for AlexNet on CIFAR-10) that the fluctuation in rank correlations between fractal dimension estimates only mildly depends on the number of SGD steps after which the dimension is estimated. To strengthen this points I encourage you to also add a column 5k vs 5k (which I expect to have similar values).
> > > >
> > > > Regarding **table 1 at 15k steps**. I was under the impression that you already had the necessary models available from the comparison of fractal dimension estimates. As this does not seem to be the case, and as the compute requirements appear to be excessive, I will leave it at that.
> > > >
> > > > ---
> > > > Overall, the authors sufficiently addressed all of my (ant other reviewers') points. I will raise my score from 5 to **7: Accept**.

---

> > > > > ### Author Response · Authors · 2024-08-13
> > > > > **Reply to Reviewer garn**
> > > > >
> > > > > We are very grateful to the reviewer for their reply and are really pleased to see that they feel that we have sufficiently addressed the points raised by them and other reviewers. We would like to reiterate our appreciation for their careful review, which has been instrumental in improving the quality of our work, making the results and interpretations more nuanced and precise. Thank you for your continued support and for engaging with us in this meaningful discussion!

---

### Official Review · Reviewer_whkD · 2024-07-12

**Soundness:** 3
**Presentation:** 3
**Contribution:** 3
**Rating:** 6
**Confidence:** 1

**Summary:**

This paper performs experiments to evaluate the correlation between the generalization gap and the persistent homology (PH) dimension, a measure of fractal dimension deriving from topological data analysis. The authors identify that the $\ell^2$ norm of the final parameter iterate correlates more strongly with the generalization gap than the fractal dimension. Moreover, the authors also demonstrate that the fractal dimension fails to predict the generalization of models trained from poor initializations.

**Strengths:**

The authors provide extensive experiments and the conclusions are well supported. Moreover, this paper is well-organized and clearly written.

**Weaknesses:**

1. The definition of the PH dimension is not easy to understand.
2. Some results need more explanation (see Questions).

**Questions:**

1. In Table 1, the correlation coefficients could have different signs in the same line. Why would this happen? Does this come from different models and data?
2. In Table 2, how to explain the influence of the batch size on the p-value?
3. I cannot figure out how Figure 1 manifests double descent.

**Limitations:**

The authors discuss limitations in Section 5.

---

> ### Author Rebuttal · Authors · 2024-08-06
>
> We thank this reviewer for their feedback and are happy to see that they found our conclusions “well-supported” and the text “well-written and organized.”
>
> We would like to first address the concern regarding the clarity of the PH dimension definition. We acknowledge that the concept of fractal dimension, particularly in relation to PH, is rather involved. Given the 9-page limit, our background section may not have provided sufficient detail for readers unfamiliar with this topic. To address this, we will include an extended background section in the appendix, making our work more self-contained and providing an introduction to the intersection of work on fractal dimension and generalization properties of deep networks.
>
> Now, we would like to answer the specific questions raised.
>
> 1. *In Table 1, the correlation coefficients could have different signs in the same line. Why would this happen? Does this come from different models and data?*
>    We appreciate the reviewer highlighting this point, and would like to elaborate further to clarify it.
>    - Under closer inspection, we noticed that for models trained with CHD we were correlating with training loss minus test loss, rather than the absolute value of this difference, which is what Dupuis et al \[2023\] use. Correlating with the absolute value changes the sign for the whole rows corresponding to CHD.
>    - We have provided an updated table in the global answer and would like to further clarify that for models trained with CHD, the correlation is computed between the PH dimensions and the loss gap, whereas for models trained with MNIST, the correlation is computed between the PH dimensions and the accuracy gap as in Dupuis et al. \[2023\]*.*
>    - Having different signs in the correlations with different measures just means that the relation is direct or inverse. We have noticed in our experiments that the norm correlates positively with the CHD but negatively for the classification models. This could be due to the ranges of hyperparameters selected being more limited in the CHD experiments. We also note that the sign of the correlation flips for the LB ratio between regression and classification.
>    - Overall the important observation of this table is regarding the absolute value of the correlation coefficient and whether it is closer to 1, indicating a stronger correlation, or not.
>
> 2. *In Table 2, how to explain the influence of the batch size on the p-value?*
>    There are several things to be said about this influence.
>    * In Table 1, a significant correlation between the generalization gap and the learning rate/batch size ratio suggests potential confounding effects. This is also visible in Figures 1 and 2 from Dupuis et al. \[2023\], where different correlation patterns can be seen depending on the batch size (different markers in the plots) and learning rate (color).
>    * Previous methods to take into account this effect involved calculating the mean of the granulated Kendall rank coefficients for learning rate and batch size. To compute these granulated Kendall coefficients, the Kendall rank correlation for each fixed value of learning rate (or batch size) and seed is computed and then averaged over all possible values. While valid, this approach masks the observed distinct correlation patterns for fixed batch sizes and learning rates.
>    * To address this, we developed a partial correlation test. This test is specifically designed to assess if the correlation observed between two variables can be explained by a common correlation with a third variable. Given the different patterns of correlation differ with batch size, we felt compelled to divide amongst these to evaluate partial correlation. The same reasoning applies to the conditional independence test.
>    * Concerning the p-values for different batch sizes in Table 2 (in the updated version of the paper, the values with p \> 0.05, indicating that the observed correlation between fractal dimension and generalization is explained by a common correlation with learning rate, will be bolded): a clear trend for the Euclidean PH dimension is that this explanation holds for almost all batch sizes, particularly the larger ones. The trend for the loss-based fractal dimension is not systematic. For FCN5 \+ MNIST, it fails for larger batch sizes, whereas for FCN5 \+ CHD, it fails for the smaller batch sizes. The correlation is more resilient in the FCN7 models.
> 3. *I cannot figure out how Figure 1 manifests double descent.*
>    We acknowledge that the double descent behavior was not very clear from the previous plots. The reason for this is that we had to slightly modify the original procedure  from Nakkiran et al. \[2021\] to avoid learning rate annealing and batch normalization, as prescribed by the assumptions  of Dupuis et al. \[2023\] and Birdal et al. \[2021\], which involve training with a constant learning rate and without batch normalization. These changes made the double descent, already a difficult to produce effect, less clear. We managed to produce a clearer plot by re-running these experiments adjusting the constant learning rate to 0.01, which is included in the global answer. The new plots clearly show double descent manifesting in test accuracy and Euclidean PH dimension.

---

> > ### Comment · Reviewer_whkD · 2024-08-12
> >
> > Thank you for your detailed response. I keep my score.

---

> > > ### Author Response · Authors · 2024-08-13
> > > **Reply to Official Comment by Reviewer whkD**
> > >
> > > We sincerely thank the reviewer for their response and are pleased to know that our answers have addressed their questions. We are also very grateful for the time and effort they dedicated to providing valuable feedback, which has contributed to improving our work.

---

### Official Review · Reviewer_rTx2 · 2024-07-13

**Soundness:** 2
**Presentation:** 3
**Contribution:** 3
**Rating:** 5
**Confidence:** 4

**Summary:**

The paper investigates the effectiveness of using fractal dimensions, particularly the Hausdorff dimension and persistent homology dimension, as measures for predicting the generalization gap in neural networks. It demonstrates that fractal dimensions fail to predict generalization for models trained from poor initializations and shows that the $\ell_2$ norm of the final parameter iterate correlates more strongly with the generalization gap than fractal dimensions.
Experimental results show that persistent homology dimension fails to predict the generalization of models with adversarial initialization, which generalize poorly.

**Strengths:**

The paper critically evaluates the use of fractal dimensions, demonstrating that they fail to predict generalization for models trained from poor initializations. The authors conduct a rigorous statistical analysis, including partial correlation measures conditioned on network hyperparameters, revealing that fractal dimensions are not statistically significant predictors of the generalization gap when hyperparameters are considered.
The study uncovers an interesting manifestation of model-wise double descent in persistent homology-based generalization measures, suggesting potential for further research.

**Weaknesses:**

Lack of theorical results.

**Questions:**

No questions.

**Limitations:**

Computational limitations and lack of Identifiable Patterns in the Correlation Failure

---

> ### Author Rebuttal · Authors · 2024-08-06
>
> We thank the reviewer for their time and feedback on our work. We are pleased to see that they found our evaluation “critical;” our statistical analysis “rigorous”, and our demonstration of the shortcomings of the existing theory “convincing”. We also appreciate the recognition of the opportunities for further investigation stemming from our experiments.
>
> While it was challenging to identify concrete areas for improvement given the reviewer did not provide any detailed questions or requests, we would like to address their main point of criticism, namely, the lack of theoretical results in our paper.
>
> Indeed, our work is experimental: it makes the important contribution of uncovering new, negative insights on an existing theory, which has gained considerable traction in high profile outlets since its inception (according to Google Scholar, around 100 citations in total). We believe that scrutinizing existing theories, especially in rapidly evolving fields like ML, is crucial to ensure that subsequent knowledge is built on a solid foundation. The experimental focus of our approach is justified by three main points:
>
> 1. **The nature of the problem is partly empirical**, with the theory making predictions that can be directly tested. By replicating the experiments and developing further tests, as we have done in our paper, we provide clear and actionable insights into the theory’s applicability and limitations.
> 2. Our findings provide **strong empirical evidence that challenges the current understanding**. A robust experimental foundation is necessary to guide theoretical advancements, which might otherwise lack practical relevance or be misdirected.
> 3. As the reviewer noted, our results open up new opportunities for further research. We believe the **empirical evidence** presented in our paper **could inspire and inform subsequent theoretical work**, rectifying existing results or developing new ones.
>
> That being said, we understand that our presentation may have created an expectation of new theoretical results. To rectify this, we are open to updating the abstract to emphasize the empirical and experimental nature of our study, as also suggested by another reviewer.
>
> We thank the reviewer once again for their consideration and ask to provide concrete questions we can address in our rebuttal.

---

> > ### Comment · Reviewer_rTx2 · 2024-08-13
> >
> > Thanks for your response!

---

### Author Rebuttal · Authors · 2024-08-06

We thank all the referees for the careful and detailed reviews, and for their encouraging words, describing our contribution as “critical”, “rigorous” and “convincing” (rTx2); “well-supported” and “well-written and organized” (whkD); “necessary in order to establish a solid foundation future work can build upon ” (garn); or “insightful”, “useful” and “relevant” to “ML” and the “wider community” (d6bB).

The remarkable observations received have made us reflect on our work and improve it. We are attaching a 1-page document, as explained in the guidelines for rebuttals, where we include several figures that we have produced motivated by these, which we think could raise the quality of this contribution, and which we would like to explain.

- **Table 1 and new plots showing the correlations:** The reviews received by whkD and garn made us decide to make several adjustments, so that their concerns could be solved and comparability with the corresponding tables in Dupuis et al \[2023\] were clearer. We are now attaching the new table; the main differences being:
  - On further inspection of the code for Dupuis et al. [2023], we realize that the absolute loss gap was computed and used for the correlation despite the definition of signed loss gap. We now recompute affected correlations, for evaluations from the regression models on CHD, to align more with the previous evaluation.
  - We have dropped the step size from the table as it was not used in our analysis any further.
  - We have differentiated between generalization measures (PH dims and norm) and the learning rate/batch size ratio. The latter is based on hyperparameters set a priori, not a computable quantity from training. Therefore, its nature and role is quite different from the former measures. We include it in the table since the very high correlations signal the potential confounding effects of this parameter on the other correlations observed, which were the parting point and inspiration for the subsequent tests we implemented (partial correlation and conditional independence).
  - Previously we computed the correlation coefficients using computed measures over all seeds. In Dupuis et al. \[2023\] the correlations were computed for each seed, then the average and standard deviation presented. We now also implemented this approach, obtaining more similar results and thus solving a concern raised by reviewer garn.
  - A significant difference that must be addressed with respect to the results presented in Dupuis et al. \[2023\] is in the case of Alexnet \+ CIFAR10, where we obtained negative correlations for the loss based measure. This is due to the appearance of some points with high learning rate close to the y-axis with very high PH dimension. After long consideration, we have been unable to determine why these points appear in our experiments but not in prior studies.  What we can say is that they do not seem to be spurious, and that they should be taken into account as they achieve 100% train accuracy, indicating that the limiting distribution has likely been reached.
- **New plot for the adversarial initialization with more seeds:** Motivated by a question from garn, we extended our experiments with adversarial initialization to 30 (instead of 10\) seeds per model. We also provide a clearer presentation of these results with Figure 4 in the attached file. There we can see how the poorly generalizing models, with higher accuracy gaps, obtained from adversarial initialization labeled by the red squares have lower PH dimension for the CNN \+ CIFAR-10 and FCN-5 \+ MNIST models than the standard ones. This observation contradicts the proposed theory, as lower fractal dimension should imply better generalization properties, i.e. lower accuracy gap.
- **New plot for the double descent:** We have been able to produce the double descent experiment more clearly using a higher learning rate of 0.01. For this plot we now use the mean of 3 seeds. Double descent clearly manifests in evaluation accuracy and Euclidean PH dimension.The behavior of loss-based ph dimension does not follow the double descent pattern, with notable instability / variance in the region of the evaluation accuracy double descent trough.
- **Table with experiments of correlation suggested by reviewer garn** to evaluate the estimations of the PH dimensions. We extended the SGD run by 5k, 10k, and 15k additional iterations and computed the PH dimensions at these steps for 10 different seeds of 3 models. We found similar magnitudes and significant rank correlations between the computations at 5k and 10k iterations, as well as at 10k and 15k iterations.

Now, three proposed improvements motivated by the reviews received:

- **Clearer setting of the hypothesis in the non-parametric permutation tests** to report statistical significance, motivated by the insightful comments of d6bB regarding the partial correlation and conditional mutual information results.
- From the comments by whkD and d6bB, we would like to add **more background to the supplementary materia**l to make the contribution self contained and accessible to any reader unfamiliar to this topic.
- Improving the **appearance of tables** and adding **more interpretations** as asked by reviewers garn and d6bB, and inspired by our answers to their careful observations.

---

### Decision · Program_Chairs · 2024-09-25

**Decision:**

Accept (poster)

**Comment:**

All the reviewers agreed that the paper has significant contributions in a growing literature on fractal-based analysis of optimization algorithms. Several concerns have been raised during the review/rebuttal period and the authors have alleviated them (at least partially) by their clarifications, preliminary results and many promises that would be implemented in the next version.

Given the current status of the paper I am recommending an acceptance by trusting the authors that they will properly implement all the promised changes one by one.